# ANOMALY DETECTION FOR TABULAR DATA WITH INTERNAL CONTRASTIVE LEARNING

**Tom Shenkar & Lior Wolf**
Blavatnik School of Computer Science, Tel Aviv University
`tomshenkar@gmail.com,wolf@cs.tau.ac.il`

## ABSTRACT

We consider the task of finding out-of-class samples in tabular data, where little can be assumed on the structure of the data. In order to capture the structure of the samples of the single training class, we learn mappings that maximize the mutual information between each sample and the part that is masked out. The mappings are learned by employing a contrastive loss, which considers only one sample at a time. Once learned, we can score a test sample by measuring whether the learned mappings lead to a small contrastive loss using the masked parts of this sample. Our experiments show that our method leads by a sizable accuracy gap in comparison to the literature and that the same default rule of hyperparameters selection provides state-of-the-art results across benchmarks.

## 1 INTRODUCTION

In the one-class classification problem, one learns a model with the goal of identifying whether a test sample belongs to the same distribution from which the training set is sampled (Schölkopf et al., 1999). Methods for solving this problem, therefore, define a criterion that is satisfied for the samples of the training set, while being less likely to hold for samples from other (unseen) distributions.

When considering perceptual data, one can rely on the structure of the input. For example, images can be rotated, and the discrimination between the various rotations is class-dependent and, therefore, indicative of the class (Golan & El-Yaniv, 2018). In this work, we consider tabular data, in which there is no prior information on the structure of the data.

If one assumes that no such structure exists, i.e., that the variables are independent, then the criterion is defined by combining the per-feature scores. This, however, is not competitive in cases in which the features in each sample vector are not independent (Ahirwar et al., 2012). By making the assumption that the dependency structures are class-dependent, one can construct, for example, a low-dimensional subspace and expect out-of-distribution classes to lie outside it (Jolliffe, 1986).

In this work, we make the assumption that the way in which a subset of the variables in the feature vector is related to the rest of the variables is class-dependent. Our method considers subsets of consecutive variables. In this manner, for a given input sample $x_i \in \mathbb{R}^d$, we obtain a set of pairs $\{(a_i^j, b_i^j)\}_{j=1}^m$, where $a_i^j$ is a vector of $k$ consecutive features from $x_i$, $b_i^j$ is the vector of all other feature values, and $m = d - k + 1$. All vectors $a_i^j$ have the same length and vary according to the first coordinate of $x_i$, from which the subsets are collected.

Given a training set, we learn a neural mapping $F : \mathbb{R}^{d-k} \to \mathbb{R}^u$ for the vectors of type $b$ and a mapping $G : \mathbb{R}^k \to \mathbb{R}^u$ for vectors of type $a$ such that the mutual information in the latent space of dimensionality $u$ between matching elements $(a_i^j, b_i^j)$ is maximized. The same networks $F, G$ are learned for all samples $i$ of the training set and for all starting indices $j$. The maximization of the mutual information is done through contrastive learning (Oord et al., 2018).

At test time, the anomaly score that is being used $y(x)$ is exactly the loss that is being minimized during training. This puts our method on solid ground, in contrast to many other outlier detection methods, in which networks are trained for one goal, and anomaly identification assumes that some regularity is learned. Additionally, since the method employs a sliding window approach, it is

straightforward to identify the specific features in the input vector that contribute to a high anomaly score. This provides our method with a natural and direct interpretability.

**Assumptions and design choices** Our method is generic and assumes little on the structure of the data. We make the following set of assumptions and associated design choices. First, we assume that one can learn networks $F, G$ that identify matching elements $(a_i^j, b_i^j)$. Second, we assume that when training $F, G$ on data from a single class, the learned networks are class-specific in the sense that the recognition rate for matching elements that belong to samples from other classes would be considerably lower. Both of these assumptions are validated in Appendix A.

Note that the contrastive learning task that $F$ and $G$ are trained to solve can be easily solved with a very short deterministic program that is class-independent, which checks the overlap between $b_i^j$ and $a_i^j$. However, the success of our method implies that by training our neural method, one learns class-specific models. Furthermore, simulations on synthetic data demonstrate that learning the class-independent solution requires a lengthy training process and does not converge to a perfect classifier. These results, which shed light on self-supervised training, are given in Sec. 5.

Our assumptions are further validated by an extensive set of experiments, in which our method, using a single architecture and the same rule of hyperparameters selection, obtains state-of-the-art results in one-class classification of tabular data. The increase in accuracy compared to existing methods is sizable and consistent across benchmarks. It is further shown that the method is insensitive to its hyperparameters, demonstrating that the validity of the assumption is a stable phenomenon. Even with $k = 1$ (masking of a single feature), the method outperforms all baselines by a gap.

Two softer assumptions are introduced by performing specific design choices (first soft assumption) or implementing the method in a specific way (second). (i) For simplicity, $F, G$ can learn to match $a_i^j$ with $b_i^j$ regardless of $j$. In an alternative implementation, one can learn a pair of networks $F^j, G^j$ for every $j$ or employ positional encoding for $j$. Since our method is effective with a single pair of networks, this option was not explored. (ii) The original order in which the features are given is used, based on the intuition that in several datasets, the order is not random. Since our method, by default, considers consecutive vector elements, nearby features are stacked together. Naturally, the strength of this effect is dataset-dependent. We quantify it in Sec. 4, and show, by gathering statistics over a large corpus of varied datasets, that the original order does not provide a clear advantage over a random order of features (consecutive features after applying permutation results in random, non-overlapping, subsets of features). This is also evident from the results presented in Appendix A.

## 2 RELATED WORK

The main application of one-class classification methods is **anomaly detection**; that is, identifying outliers after observing a set of mostly normal (the opposite of abnormal) samples (Chandola et al., 2009; Pang et al., 2020; Ruff et al., 2021). A straightforward way to perform this task is to model a distribution based on the training samples and then estimate the likelihood of each test sample. For this purpose, one can employ non-parametric methods, such as kernel density estimation (Parzen, 1962), or the very recent COPOD method (Li et al., 2020) that is based on an empirical copula model. Parametric methods include Gaussian and Gaussian mixture models (Zong et al., 2018), as well as adversarial learning (Schlegl et al., 2017). An alternative to density estimation approaches relies on regularized classifiers. The classical methods were mostly kernel-based methods, in which the role of the regularization term is to ensure that the fitted model is tight around the observed samples (Schölkopf et al., 1999). Many of the first deep learning anomaly detection methods employed such classical one-class methods on top of auto-encoder based representations (Hawkins et al., 2002; Sakurada & Yairi, 2014; Xia et al., 2015; Xu et al., 2015b; Erfani et al., 2016; An & Cho, 2015). More recent methods apply a suitable one-class loss, in order to learn a neural network-based representation in an end-to-end manner (Ruff et al., 2018). In order to further avoid the problem of representation collapse, the DROCC method (Goyal et al., 2020) applies virtual adversarial training (Miyato et al., 2018) to create virtual negative samples around the training samples.

**Self-supervised learning**, in which an unsupervised learning problem is turned into a discriminative learning problem, was introduced to anomaly detection by Golan & El-Yaniv (2018). Their method predicts the image transformation that is applied to an image. Assuming that this classification problem is class-dependent, the membership score is based on the success of the learned classifier

on a given test image. GOAD (Bergman & Hoshen, 2020) improved this method, by learning an embedding space, in which the classifier considers the distances to the center of the set of training samples after applying each transformation. The method is also made suitable for tabular data, in which case random linear projections replace the geometric transformations. Our work is based on a different self-supervised task called masking, in which part of the data is held out and is predicted by the rest of the data. This form of self supervision is commonly used for learning representation in NLP (Mikolov et al., 2013) and in computer vision (Pathak et al., 2016). As far as we can ascertain, masking was not used for one-class classification before.

The idea of **contrastive learning** has emerged in metric learning, where it was used to train a Siamese network (Chopra et al., 2005). However, its main application is in unsupervised representation learning (Hadsell et al., 2006). The learned embedding brings associated samples closer together, while pushing away other samples. The framework of noise contrastive estimation (Gutmann & Hyvärinen, 2010) casts this type of learning as a form of mutual information maximization. Many of the most recent contrastive learning methods perform unsupervised learning by anchoring an image together with its transformed version, while distancing other images (He et al., 2020; Misra & van der Maaten, 2019; Chen et al., 2020).

Recently, the contrastive learning method of Chen et al. (Chen et al., 2020) was adapted for the problem of one-class classification (Tack et al., 2020). The obtained score combines the norm of the representation (Taigman et al., 2015) with the the maximal similarity to any sample of the training set to define an anomaly score. The performance is further enhanced by contrasting two sets of image transformations: those that maintain the same-identity property vs. those that lead to a different training identity. Another method applies contrastive learning that is tailored to one-class learning in order to learn an image representation, followed by a one class classifier (Sohn et al., 2021). These image transformation-based techniques are not applicable to tabular data, since there is no group of transformations that the content of generic vectors is invariant to.

The contemporaneous NeuTraL AD work by Qiu et al. (2021) employs per-sample contrastive loss for identifying anomalies in tabular data, similar to our work. However, there are crucial differences: (1) NeuTraL AD learns specific masks, while we apply the entire set of the masks specified by a window size $k$. (2) The role of NeuTraL AD masks is to mask-out parts that are irrelevant for specific classes. In our case, we perform a two sided matching that identifies the masked part form the original. (3) NeuTraL AD learns a single feature extractor ("encoder") for both the original and transformed data. In our case, the two sides of the contrastive loss are of very different dimensions ($d - k$ and $k$) and we employ two different encoders. Furthermore, we explore in our ablation (and theoretically motivate) the importance of using encoders that are of different architectures. (4) Their method is built such that there is diversity between the different views and similarity between each view and the original. This requires the views to be spread, in the latent space, around the original sample. In our case, the constraints are such that the crop matches the complement elements of the vector more than all other crops. There is no requirement that the other crops are dissimilar between themselves.

## 3  METHOD

We are given a training set of $n$ in-class samples $S = \{x_i\}$, each a vector of $d$ dimensions. Our goal is to design a score $y : \mathbb{R}^d \to \mathbb{R}$ that maps samples from the sample domain to a low value if they are sampled from the underlying distribution from which $S$ is sampled and to a high one otherwise.

The method has two hyperparameters that specify dimensions: $k < d$ determines the size of the subset of features we consider, and $u$ determines their embedding size. A third hyperparameter $\tau$ is the temperature constant of the loss.

The method first constructs a set of $m = d + 1 - k$ pairs $\Phi(x_i) = \{(a_i^j, b_i^j)\}$ from each training sample $x_i$. Each pair in this set is obtained by extracting $k$ consecutive variables from $x_i$. Let $a_i^j$, $1 \le j \le m$ be the vector $[x_i^j, x_i^{j+1}, \ldots, x_i^{j+k-1}]$, where superscripts denote elements of the vector $x_i$. We define $b_i^j = [x_i^1, x_i^2, \ldots, x_i^{j-1}, x_i^{j+k}, \ldots, x_i^d]$ to be the vector of the other $d - k$ elements in $x_i$. The method then learns two mappings $F, G$ that maximize the mutual information between $F(b_i^j)$ and $G(a_i^j)$, where $(a_i^j, b_i^j) \in \Phi(x_i)$, $i = 1 \ldots n$. The same mappings are learned for all samples and regardless of the index $j \in [1, m]$.

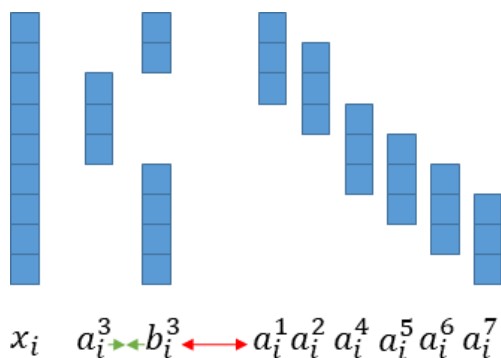

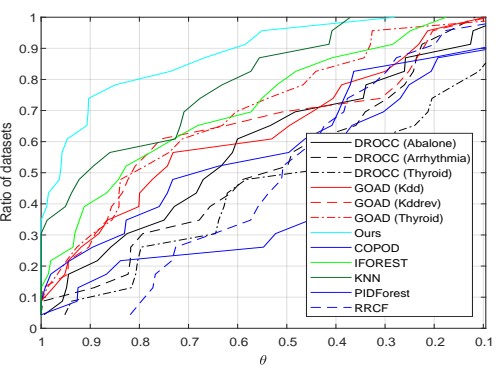

Figure 1: The underlying learning problem. Given a sample vector $x_i$, we consider the subvector $a_i^3$ and its complementary $b_i^3$. The networks are trained to produce similar embeddings for this pair of vectors, while distancing the embedding of $a_i^{j'}$ for $j' \neq 3$ from that of $b_i^3$.

Figure 2: A Dolan-More profile for ODDSs F1 scores. The x-axis is a varying threshold ($\theta$). The plot considers the performance of each method in comparison to the best performing method. The y-axis is, for a given method, the ratio of datasets in which the obtained score is above $\theta$ times the maximal score over all methods. Experiments with missing baselines were omitted for fairness.

The mutual information is maximized through the use of the noise contrastive estimation framework (Oord et al., 2018), see Fig. 1 for an illustration of the contrastive relations in our setting. In this framework, there is a query $q$, a positive sample $v^+$, and negative ones $v^-$, all vectors in $\mathbb{R}^u$. Contrastive learning maximizes the similarity of the query with the positive sample, while minimizing it with the negative samples. In our case, the vectors are given, for some $i, j, j' \neq j$ as:
$$q = F(b_i^j), v^+ = G(a_i^j), v^- = G(a_i^{j'}).$$

Almost all current contrastive learning methods employ normalization of the vectors $q$ and $v^+, v^-$ such that their unit norm is 1. In our method, this normalization is performed after a first normalization step that is applied at each vector dimension in $\mathbb{R}^u$ separately, by considering all of the sub-vectors of the input vector $x_i$. The motivation for the first normalization step is that scale, by itself, can lead to the identification of a certain feature. This scaling of the features is often class-independent, since it frequently depends more on the nature of the feature than on the class, while the classification task is required to be class-dependent.

The normalized network $F^N$ considers the $u \times m$ matrix $B = [F(b_i^1), F(b_i^2), \ldots, F(b_i^m)]$ and normalizes each of its rows to have an L2 norm of 1 to obtain a matrix $B^N$. We define the normalized network $F^N$ such that $F^N(b_i^j)$ is the $j$−th column $B^N$, further normalized to an L2-norm of one. Although we omit it from the operand list, $F^N(b_i^j)$ depends not only on $b_i^j$ but on all $b$-type vectors in $\Phi(x_i)$ through normalization. Similarly, $G^N(a_i^j)$ is defined by the matrix $A^N$, which is a double-normalized version of the matrix that contains vectors of the form $G(a_i^j)$, where $j$ varies by column.

The contrastive loss $\ell$ is defined as an $m$-way classification problem, in which the cross-entropy loss for a given temperature $\tau$ is used. The logit used is the pseudo-probability for the positive sample $v^+$ being selected over $m - 1$ negatives given the query $q$ (Wu et al., 2018; He et al., 2020). Note that the normalized versions of these vectors are being used.

$$\ell(F, G, \Phi(x_i), j) = -\ln \frac{\exp(F^N(b_i^j) \cdot G^N(a_i^j)/\tau)}{\sum_{j'=1}^{m} \exp(F^N(b_i^j) \cdot G^N(a_i^{j'})/\tau)}. \tag{1}$$

Once the networks $F$ and $G$ are trained with the contrastive loss, we define the one-class classification score as $y(x) = \sum_j \ell(F, G, \Phi(x), j)$, where $\Phi(x)$ is constructed for sample $x$ similarly to the construction of $\Phi(x_i)$ for training sample $x_i$.

### TRAINING, PARAMETERS, AND ARCHITECTURE

During training, the method directly minimizes the anomaly score of the training set $S$, given by $\mathcal{L} = \sum_{x \in S} y(x)$. Training employs the Adam optimizer (Kingma & Ba, 2014) with a learning rate

of $10^{-3}$ . It stops when the loss is smaller than $10^{-3}$ for datasets with $d < 40$. For larger input dimensions, this criterion would lead to long training sessions, and we use $10^{-2}$ instead.

We fix $\tau = 0.01$ and $u = 200$. The value of $u$, which is often much larger than $k$, provides enough capacity throughout all experiments, without the need to tune it for each problem. We set $k$ proportionally to the input dimension $d$. For $d$ smaller than 40, we set $k = 2$, for $d$ in the range $[40, 160]$ we employ $k = 10$, and for $d > 160$, $k$ takes the value $d - 150$. However, as our experiments in Sec. 4 show, the method is not very sensitive to the value of $k$.

We found that we can make use of the fact that the features are unordered and simply combine multiple scores, each obtained on a different permutation of the features. In our experiments (Sec. 4), repeating this way is only very seldom detrimental to accuracy, and it improves performance for small $d$ and very small $n$. On the other hand, it does add to the overall runtime.

In order to make use of this bagging effect, when needed, we set the number of repeats to be $r = 1 + \lfloor 100(\log(n) + d)^{-1} \rfloor$. For each repeat after the first, we randomly permute the set of features. The score that the method returns is the mean of the scores obtained from each such repeat.

$F$ and $G$ are two fully connected networks with LeakyRELU activations (Xu et al., 2015a) using a slope coefficient of 0.2 in all layers, except for the first layer of $F$, which has a $tanh$ activation. The reason is that we wish to distance the embedding of the $a$-part of $x$ and its $b$ part, making a simple matching between the parts, which overlap for $b_i^j$ and $a_i^{j'}$ when $j' \neq j$, more challenging. See Sec. 5 for a detailed discussion. $F$ has two hidden layers, with $u$ and $2u$ hidden units, each followed by Batch Normalization (Ioffe & Szegedy, 2015). $G$ is similar, only that due to the smaller input sizes, the hidden layers have $u/4$ and $u/2$ units and Batch Normalization is applied only after the first layer.

## 4 EXPERIMENTS

**Datasets** Based on the terminology of the field of anomaly detection, we use the term "normal" to describe the class observed during training, and abnormal to describe samples from the other class or classes. The experiments were conducted on two groups of datasets: (i) a collection of four datasets commonly used for reporting anomaly detection for tabular data, (ii) a much more comprehensive set of tabular datasets for benchmarking outlier detection. The first set of datasets contains two small-scale medical datasets (Arrhythmia and Thyroid), as well as two cyber-intrusion detection datasets (KDD and KDDRev) which are considerably larger. Following Zong et al. (2018) and others, the categorical attributes are presented to the network as one-hot vectors. The second set employs the "Multi-dimensional point datasets" from the Outlier Detection DataSets (ODDS)[1]. It contains 31 datasets, including two of the four datasets above. The datasets Heart and Yeast were omitted, as it is not clear from the description which class is the normal one. See App. B the dimensionality data.

**Evaluation protocol and scores** Following Zong et al. (2018); Bergman & Hoshen (2020), the training set contains a random subset of 50% of the normal data. The test set contains the rest of the normal data, as well as all the anomalies. In the first set of experiments, the mean and standard deviation (SD) of the F1 score are reported. For the experiments we run, this is computed over 500 random splits for the smaller datasets (Arrhythmia and Thyroid) and 10 splits for the larger ones (KDD and KDD-Rev). For the baseline methods, the sample size varies and in some cases the results reported in the literature are given without SD. Following the existing protocol, the decision threshold for scoring the methods is chosen such that the number of test samples above this threshold (i.e., classified as anomalies) is the number of anomalies in the test set. For the second set of experiments, in which all runs were made by us, we report (sometimes in the appendix), in addition to F1, the AUC. Since the AUC varies less dramatically than F1 scores, it is more suitable for comparison across datasets. It also has the advantage of not requiring setting a threshold.

**Baseline methods** For the first set of experiments, a comprehensive set of literature baselines is presented. One-Class SVM (OC-SVM) (Schölkopf et al., 1999)) and DAGMM Zong et al. (2018) results are reported as computed by Zong et al. (2018). GOAD (Bergman & Hoshen, 2020), Local Outlier Factor (LOF) (Breunig et al., 2000), and an ensemble method (FB-AE) that employs autoencoders as the base classifier, feature bagging as the source of randomization, and reconstruction error as the anomaly score, are reported by Bergman & Hoshen (2020). A recent baseline, DROCC Goyal

---

[1] `http://odds.cs.stonybrook.edu/`, accessed January 2021

Table 1: Mean F1 (percents) and standard deviation over multiple resampling on the four datasets commonly used in the literature. [1]DROCC is reported based on our runs due to protocol discrepancies in the published code; some experiments are missing due to limitations of the published code.

| | Arrhythmia | | Thyroid | | KDD | | KDDRev | |
|---|---|---|---|---|---|---|---|---|
| Method | $F_1$ Score | SD | $F_1$ Score | SD | $F_1$ Score | SD | $F_1$ Score | SD |
| OC-SVM (Schölkopf et al., 1999) | 45.8 | | 38.9 | | 79.5 | | 83.2 | |
| LOF (Breunig et al., 2000) | 50.0 | 0.0 | 52.7 | 0.0 | 83.8 | 5.2 | 81.6 | 3.6 |
| DAGMM (Zong et al., 2018) | 49.8 | | 47.8 | | 93.7 | | 93.8 | |
| FB-AE (Chen et al., 2017) | 51.5 | 1.6 | 75.0 | 0.8 | 92.7 | 0.3 | 95.9 | 0.4 |
| DROCC (Goyal et al., 2020)[1] | 38.8 | 6.2 | 72.7 | 3.1 | N/A | N/A | N/A | N/A |
| GOAD (Bergman & Hoshen, 2020) | 52.0 | 2.3 | 74.5 | 1.1 | 98.4 | 0.2 | 98.9 | 0.3 |
| COPOD (Li et al., 2020) | 58.2 | 1.4 | 30.8 | 0.5 | 44.5 | 0.1 | 30.9 | 0.4 |
| Ours | **61.8** | 1.8 | **76.8** | 1.2 | **99.4** | 0.1 | **99.2** | 0.3 |

et al. (2020), is rerun by us based on their code, since the published results sampled the test set using a different protocol[2]. Finally, we include the COPOD baseline (Li et al., 2020), based on our runs, as a modern non-deep-learning method..

For the second set of experiments, we focus on the most recent deep methods: GOAD and DROCC. Since GOAD uses three different architectures in their code, we report results for all three. The first architecture is the one used by Bergman & Hoshen (2020) for the small datasets, the second is used for KDD, and the third is the one used for KDDRev. Similarly, DROCC (Goyal et al., 2020) employs three architectures in their code: one for Thyroid, one for Arrhythmia, and one of Abalone, and we run all three. Our method employs the same architecture in all experiments, except that $k$ is adjusted according to $d$, see Sec. 3. As non-deep baselines, we run multiple methods including classical methods and modern ones: IForest (Liu et al., 2008), k-nearest neighbours (KNN), PIDForest (Gopalan et al., 2019), RRCF (Guha et al., 2016), and COPOD (Li et al., 2020). All of these are based on our own runs, using the PyOD anomaly detection python package Zhao et al. (2019) for KNN and IForest, RRCF python package and the official code published by the authors of PIDForest in Github. The defualt parameters were used, except for the suprisignly strong baseline of KNN, for which we report results for the best parameters found on the test set, see Appendix C.

**Results** The results for the **first set** of experiments are reported in Tab. 1. Our method outperforms the literature baselines by a significant margin on Arrhythmia and Thyroid, where the baselines obtain a moderate F1 score. On the larger datasets - KDD and KDDRev, where the performance of GOAD is very high - we outperform it, obtaining a near-perfect score. On all datasets, the pvalue of the Wilcoxon test between our method and the best competing method is lower than 0.001. From the baseline methods, the feature bagging auto encoder (FB-AE) and GOAD seem to be the strongest, while DROCC, with the correct protocol, is not as competitive. COPOD, despite being shown to be successful on many other benchmarks, does not perform particularly well on this first set.

The results for the **second set** of experiments are obtained by our runs and compared with both deep and classical methods (we separate these into two tables due to layout considerations). The deep baselines, GOAD and DROCC, are run each with three different architectures. The mean and SD for the F1 score across 20 runs are reported in Tab. 2, see Appendix C for the AUC results.

A similar experiment is conducted in comparison to classical methods: IForest (Liu et al., 2008), KNN, PIDForest (Gopalan et al., 2019), RRCF (Guha et al., 2016), and COPOD (Li et al., 2020). The latter has been shown to be highly effective on the ODDS collection, when tested with a random 60%/40% train/test split protocol (Li et al., 2020).

As can be seen, in Tab. 2,3, our method obtains the highest mean performance. The mean rank reported in the tables considers the results of both tables together, and our mean rank is considerably higher than all classical and deep baselines. KNN, perhaps the simplest method, obtains the second

---

[2]In their implementation, (1) the number of normals in the test set equals the number of anomalies in the data, and the rest are used for training and (2) the threshold in test time to determine F1 is constant across datasets at 20% of the *test* set.

Table 2: F1 for the ODDS benchmarks for deep models. To avoid selecting the architecture for the baseline methods, we report all versions. Missing experiments are due to the limitations of the published code.

| Method | DROCC (Thyroid) | DROCC (Arrhythmia) | DROCC (Abalone) | GOAD (Thyroid) | GOAD (kddrev) | GOAD (kdd) | Ours |
|---|---|---|---|---|---|---|---|
| Wine | 20.0±19.0 | 32.0±35.4 | 63.0±20.0 | 67.0±9.4 | 76.0±10.8 | 42.2±26.9 | **90.0**±6.3 |
| Lympho | 0.0±0.0 | 38.3±23.6 | 65.0±5.0 | 68.3±13.0 | 67.7±7.8 | 46.0±21.5 | **86.7**±6.0 |
| Glass | 22.2±17.2 | 13.3±12.0 | 14.5±11.1 | 12.7±3.9 | 25.7±12.0 | 24.0±15.1 | **27.2**±10.6 |
| Vertebral | 25.7±5.4 | 27.0±15.9 | 9.3±6.1 | 16.3±9.6 | 26.9±5.2 | 25.5±4.7 | **26.0**±7.7 |
| Wbc | 0.0±0.0 | 18.6±16.0 | 9.0±6.2 | 66.2±2.9 | 16.8±16.1 | 57.2±6.9 | **67.6**±3.6 |
| Ecoli | N/A | N/A | N/A | 61.4±31.7 | 69.3±23.7 | 66.1±27.8 | **70.0**±7.8 |
| Ionosphere | 29.9±6.8 | 76.3±6.4 | 76.9±2.8 | 83.4±2.6 | 88.1±2.3 | 88.7±2.7 | **93.2**±1.3 |
| Arrhythmia | 38.8±6.2 | 37.9±8.0 | 37.1±6.8 | 52.0±2.3 | 51.6±4.0 | 45.2±7.6 | **61.8**±1.8 |
| Breastw | 15.3±7.7 | 63.8±29.3 | 93.0±3.7 | 96.0±0.6 | 73.5±9.4 | 94.8±1.0 | **96.1**±0.7 |
| Pima | 40.6±3.3 | 55.2±8.0 | **66.0**±4.1 | **66.0**±3.1 | 57.3±1.9 | 60.2±2.0 | 59.1±2.2 |
| Vowels | 33.0±16.4 | 20.4±15.0 | 66.2±8.8 | 31.1±4.2 | 78.6±2.9 | 72.6±4.5 | **90.8**±1.6 |
| Letter | 39.0±4.8 | 31.3±6.5 | 55.6±3.6 | 20.7±1.7 | 53.8±2.2 | 48.6±3.0 | **62.8**±2.4 |
| Cardio | 62.6±6.1 | 53.3±12.9 | 49.8±3.2 | **78.6**±2.5 | 48.9±5.8 | 58.4±4.8 | 71.0±2.4 |
| Seismic | 17.7±2.5 | 17.9±2.7 | 19.1±0.9 | **24.1**±1.0 | 18.6±1.9 | 19.4±2.6 | 20.7±1.9 |
| Musk | 1.3±3.3 | 99.7±0.9 | 99.4±1.5 | **100.0**±0.0 | **100.0**±0.0 | **100.0**±0.0 | **100.0**±0.0 |
| Speech | 3.4±2.4 | 2.1±1.9 | 4.3±2.0 | 4.8±2.3 | 8.9±2.9 | 4.4±2.4 | **5.2**±1.2 |
| Thyroid | 68.4±3.2 | 69.7±5.7 | 72.7±3.1 | 72.5±2.8 | 17.2±9.4 | 32.9±9.9 | **76.8**±1.2 |
| Abalone | 44.3±17.6 | 11.6±10.5 | 17.9±1.3 | 57.6±2.2 | 6.2±1.4 | 6.6±1.0 | **68.7**±2.3 |
| Optidigits | 18.4±5.4 | 26.5±12.6 | 30.5±5.2 | 0.3±0.3 | 45.8±2.6 | 36.5±9.9 | **66.3**±10.1 |
| Satimage-2 | 10.2±2.5 | 33.7±19.6 | 4.8±1.6 | 90.7±0.7 | 20.4±10.5 | 21.7±2.2 | **92.4**±0.7 |
| Satellite | 61.3±6.3 | 68.1±0.7 | 52.2±1.5 | 64.2±0.4 | 67.9±2.0 | 70.1±0.8 | **73.2**±1.6 |
| Pendigits | 7.9±2.9 | 10.6±7.9 | 11.0±2.6 | 40.1±5.0 | 25.1±3.6 | 19.4±4.5 | **82.3**±4.5 |
| Annthyroid | 63.8±4.7 | 55.6±5.2 | 64.2±3.3 | 50.3±6.3 | 61.4±7.8 | **68.0**±3.7 | 45.4±1.8 |
| Mnist | N/A | N/A | N/A | 66.9±1.3 | 67.5±1.2 | 66.2±1.5 | **85.9**±0.0 |
| Mammo. | 34.1±2.2 | 31.5±6.2 | 32.6±2.1 | **33.7**±6.1 | 16.5±1.3 | 16.0±1.5 | 29.4±1.4 |
| Shuttle | N/A | N/A | N/A | 73.5±5.1 | N/A | **98.4**±0.2 | **98.4**±0.1 |
| Mullcross | N/A | N/A | N/A | 99.7±0.8 | N/A | 36.4±17.0 | **100.0**±0.0 |
| Forest | N/A | N/A | N/A | 0.1±0.2 | N/A | 15.0±4.3 | **44.0**±4.1 |
| Kdd | N/A | N/A | N/A | 79.6±3.9 | N/A | 98.4±0.2 | **99.4**±0.1 |
| Kdd-rev | N/A | N/A | N/A | 98.0±0.1 | 98.9±0.3 | 98.8±0.1 | **99.2**±0.3 |
| mean | 21.7±4.6 | 29.6±8.5 | 33.6±3.3 | 55.9±4.2 | 42.8±4.8 | 51.3±6.3 | **69.7**±2.9 |
| mean rank (out of 12) | 9.1 | 8.8 | 8.0 | 5.3 | 6.5 | 5.7 | **2.6** |

best performance. While we list the KNN results for $K = 5$, which is the best test-set parameter found, KNN outperforms the other baselines for other values of $K$ as well, see Appendix C.

For all cases in which our method achieves the highest score (not including the ties at 100%), the pvalue of the Wilcoxon test is below 0.0001 when compared to the next best method.

To further visualize these multiple-benchmark results, we employ a Dolan-More profile. In such profiles, there is a single plot per method, based on the statistics of the performance it obtains compared to the best method for each benchmark. Specifically, the ratio of benchmarks for which the method obtains up to a fraction $\theta$ of the maximal score obtained by all methods. This is plotted for $0 \leq \theta \leq 1$. A leading method is expected to obtain a ratio of 1.0 closer to $\theta = 1$, which means that it obtains, on all datasets, performance that is within a relatively narrow multiplicative margin of the best method. A dominating method would present a graph with higher values along the y-axis for all $\theta$ values, which means that for every margin, it has more datasets in which it is within margin of the best results than any other method. Fig. 2 presents our results for the F1 score, showing that our method leads by a significant margin over the 11 other alternatives.

**Interpretability** Since we apply a sliding window approach and obtain the anomaly score by simple integration, we can point back to specific features in the input vector that are overrepresented in factors that indicate the anomaly. This provides a natural explainability capability, which, as far as we can ascertain, was not shown for the deep-learning based anomaly detection methods (interpretability can be provided for such methods indirectly, e.g., by tracking network activations (Bach et al., 2015) or linear approximations (Ribeiro et al., 2016)).

Table 3: F1 results traditional models. The results are reported to value of $K$ that was found to provide the highest mean F1 score.

| Method | COPOD | IForest | KNN | PIDForest | RRCF | Ours |
|---|---|---|---|---|---|---|
| Wine | 60.0±4.5 | 64.0±12.8 | **94.0**±4.9 | 50.0±6.4 | 69.0±11.4 | 90.0±6.3 |
| Lympho | 85.0±5.0 | 71.7±7.6 | 80.0±11.7 | 70.0±0.0 | 36.7±18.0 | **86.7**±6.0 |
| Glass | 11.1±0.0 | 11.1±0.0 | 11.1±9.7 | 8.9±6.0 | 15.6±13.3 | **27.2**±10.6 |
| Vertebral | 1.7±1.7 | 13.0±3.8 | 10.0±4.5 | 12.0±5.2 | 8.0±4.8 | **26.0**±7.7 |
| Wbc | **71.4**±0.0 | 70.0±3.7 | 63.8±2.3 | 65.7±3.7 | 54.8±6.1 | 67.6±3.6 |
| Ecoli | 25.6±11.2 | 58.9±22.2 | **77.8**±3.3 | 25.6±11.2 | 28.9±11.3 | 70.0±7.8 |
| Ionosphere | 70.8±1.8 | 80.8±2.1 | 88.6±1.6 | 67.1±3.9 | 72.0±1.8 | **93.2**±1.3 |
| Arrhythmia | 58.2±1.4 | 60.9±3.3 | **61.8**±2.2 | 22.7±2.5 | 50.6±3.3 | **61.8**±1.8 |
| Breastw | 96.4±0.6 | **97.2**±0.5 | 96.0±0.7 | 70.6±7.6 | 63.0±1.8 | 96.1±0.7 |
| Pima | 62.3±1.1 | **69.6**±1.2 | 65.3±1.0 | 65.9±2.9 | 55.4±1.7 | 59.1±2.2 |
| Vowels | 4.8±1.0 | 25.8±4.7 | 64.4±3.7 | 23.2±3.2 | 18.0±4.6 | **90.8**±1.6 |
| Letter | 12.9±0.7 | 15.6±3.3 | 45.0±2.6 | 14.2±2.3 | 17.4±2.2 | **62.8**±2.4 |
| Cardio | 65.0±1.4 | **73.5**±4.1 | 67.6±0.9 | 43.0±2.5 | 43.9±2.7 | 71.0±2.4 |
| Seismic | 29.2±1.3 | **73.9**±1.5 | 30.6±1.4 | 29.2±1.6 | 24.1±3.2 | 20.7±1.9 |
| Musk | 49.6±1.2 | 52.0±15.3 | **100.0**±0.0 | 35.4±0.0 | 38.4±6.5 | **100.0**±0.0 |
| Speech | 3.3±0.0 | 4.9±1.9 | 5.1±1.0 | 2.0±1.9 | 3.9±2.8 | **5.2**±1.2 |
| Thyroid | 30.8±0.5 | **78.9**±2.7 | 57.3±1.3 | 72.0±3.2 | 31.9±4.7 | 76.8±1.2 |
| Abalone | 50.3±6.4 | 53.4±1.7 | 43.4±4.8 | 58.6±1.6 | 36.9±6.4 | **68.7**±2.3 |
| Optidigits | 3.0±0.3 | 15.8±4.3 | **90.0**±1.2 | 22.5±16.8 | 1.3±0.7 | 66.3±10.1 |
| Satimage-2 | 77.9±0.9 | 86.5±1.7 | **93.8**±1.2 | 35.5±0.4 | 47.9±3.4 | 92.4±0.7 |
| Satellite | 56.7±0.2 | 69.6±0.5 | **76.3**±0.4 | 46.9±3.7 | 55.4±1.3 | 73.2±1.6 |
| Pendigits | 34.9±0.6 | 52.1±6.4 | **91.0**±1.4 | 44.6±5.3 | 16.3±2.6 | 82.3±4.5 |
| Annthyroid | 31.5±0.5 | 57.3±1.3 | 37.8±0.6 | **65.4**±2.7 | 32.1±0.8 | 45.4±1.8 |
| Mnist | 38.5±0.4 | 51.2±2.5 | 69.4±0.9 | 32.6±5.7 | 33.5±1.7 | **85.9**±0.0 |
| Mammo. | **53.4**±0.9 | 39.0±3.3 | 38.8±1.5 | 28.1±4.3 | 27.1±1.9 | 29.4±1.4 |
| Shuttle | 96.0±0.0 | 96.4±0.8 | 97.3±0.2 | 70.7±1.0 | 32.0±2.2 | **98.4**±0.1 |
| Mullcross | 66.0±0.1 | 99.1±0.5 | **100.0**±0.0 | 67.4±2.1 | **100.0**±0.0 | **100.0**±0.0 |
| Forest | 18.2±0.2 | 11.1±1.6 | **92.1**±0.3 | 8.1±2.8 | 9.9±1.5 | 44.0±4.1 |
| Kdd | 44.5±0.1 | 95.6±2.2 | 98.9±0.4 | 92.1±2.2 | 74.7±0.9 | **99.4**±0.1 |
| Kdd-rev | 30.9±0.4 | 96.4±2.4 | 85.2±0.3 | 50.9±4.1 | 9.8±1.2 | **99.2**±0.3 |
| mean | 44.7±1.5 | 58.2±4.0 | 67.7±2.2 | 43.4±3.9 | 37.0±4.2 | **69.7**±2.9 |
| mean rank (out of 12) | 7.1 | 4.7 | 3.8 | 7.7 | 8.4 | **2.6** |

As an example, consider the "Thyroid" dataset. In the ODDS repository, the normal class contains samples labeled "normal" or "sub-normal functioning" and the anomalies are from the class "hyper-functioning". Examining the loss score per each sliding window (and not just the mean over all of them) reveals that in 83% of the anomalies detected the feature with the highest loss value (implying the lowest correlation with the rest of the vector compared with the training set) was "T3 concentration", in 12% it was "TSH levels" and the rest were "T4U concentration". This makes medical sense since hyperthyroidism is characterized by high T3 and T4U levels, and low TSH.

To examine the ability of the model to identify dependencies, we conducted a random data experiment as follows: we generated random data for vectors over $\mathbb{R}^4$ by sampling from a Gaussian distribution with means $[1, 2, 3, 4]$ and a covariance matrix that implies a high covariance of 0.85 between the features of indices 2,3, zero covariance elsewhere, and unit variance for all of the features. We also sampled random data that was identical in all parameters except for having no correlation between any of the features. The model ($k = 2, r = 1$) was trained on half the data with correlation between features and tested on the other half with the addition of all non-correlated data. The mean AUC over 10 runs was 88%, which is considerably higher than the baselines, see Appendix C (Tab. 10). More interestingly, features 2,3 were highlighted by the model as those with the highest anomaly score. This was inferred by simply aggregating the losses of the sliding windows each feature is included in, per sample. In 89% of the samples detected as anomalies, either the second or third features were

ranked highest (with almost equal frequency). This is reassuring, since the only difference between the two sampling procedures was in the dependencies between the features.

## 5 DISCUSSION

The self-supervised task we seek to solve through the learning of the mappings $F$ and $G$ is relatively simple and can be solved without learning in $O(dm)$, since it amounts to identifying whether $b_i^j$ and $a_i^{j'}$ overlap, in which case $j' \neq j$, or not, which implies that $j' = j$. Since this decision process is not class-specific, should we be surprised that the learned representations seem to be highly distinctive of class membership? Consider by way of analogy the self-supervised learning of natural images by employing a constrastive loss between an anchor image, its transformation, and another image, e.g., (Chen et al., 2020). Identifying the transformation between two images, which is no more difficult than finding out whether they are related by a transformation, is done reliably with neural networks using point matching (Sarlin et al., 2020), or using "direct methods" (DeTone et al., 2016). These methods are not class-specific. However, the representations learned by applying geometric transformations are extremely distinctive of the class.

To study this further with a random data experiment, in which the vectors $x$ are composed of $d$ independent variables sampled uniformly in $[-1, 1]$, we learn networks $F$ and $G$ and observe the success in identifying index $j$, vs. the other indices. The results for $d = 6, 15, 30$ are reported in Fig. 6. As can be seen (our method, all experiments), the loss is being reduced while training. However, the network is not able to immediately obtain perfect performance, especially for larger $d$. As mentioned in Sec. 3, the architectures of $F$ and $G$ are slightly different, based on the motivation that this would make the class-independent solution less accessible during training. As shown in Fig. 6, when replacing the architecture of $G$ to be identical to that of $F$, the class-independent learning has a lower error on most epochs. Lastly, for data where the features are different from one another the problem becomes less challenging. A second random data experiment sets the mean value of each variable uniformly in [0,1] and the data is i.i.d Gaussian with a unit variance. In this case, the network reaches the same loss in one-tenth of the number of epochs, see Appendix C (Fig. 9).

Tabular data tends to have considerably higher structural variance between datasets than perceptual data. The type of features (continuous or categorical), the number of features, and dependencies between features vary greatly from one dataset to the next. This variance makes the development of a generic anomaly detection method challenging, and we find a significant difference in performance across methods. We note, however, that our method is more stable than recent ones, and handles multiple datasets using the same architecture, except for minimal tuning directly related to the dimensionality of the data. For example, on the four datasets commonly used in the literature, the method is applied exactly the same, except for Thyroid, where due to the low input dimensionality ($d = 6$), the value of $k = 2$ was used. In contrast, GOAD, for example, has dataset-dependent stopping criteria (early stopping or 25 epochs) and employs three architectures on these four datasets.

## 6 CONCLUSIONS

We present a generic one-class classification method for tabular data. The method assumes that it is possible to identify missing features based on the rest and employs a contrastive loss for learning without any other auxiliary loss. In an extensive set of experiments, the method presents a significant advantage over existing anomaly detection methods. The method requires no tuning between different datasets and is stable with respect to its hyperparameters.

**Reproducibility Statement** The full implementation of our method and scripts for reproducing the experiments are attached as a supplementary zip file. This archive includes a README file and a list of requirements that support seamless reproducibility. The runtime statistics are reported in Appendix G. We note that our method requires modest computational resources, such as those available for free on Google Colab, further supporting widespread reproducibility. A detailed depiction of the (few) hyperparameters used is given in Sec. 3. A sensitivity analysis presented in Sec. 4 demonstrates that the method is insensitive to specific parameter values.

ACKNOWLEDGMENTS

This project has received funding from the European Research Council (ERC) under the European Union's Horizon 2020 research and innovation programme (grant ERC CoG 725974).

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

## A  VALIDATING THE BASIC ASSUMPTIONS

We validate the ability to learn to match between $(a_i^j, b_i^j)$ against options of the form $(a_i^{j'}, b_i^j)$ for $j'$ by training networks $F, G$ and measuring the log-likelihood of the correct answer (Eq. 1) for samples from the normal class. We then repeat this for out-of-class samples to validate that the learned networks are class dependent.

As can be seen in Fig. 3 and 4, for the vast majority of 'Multi-dimensional point datasets" from the Outlier Detection DataSets (ODDS, `http://odds.cs.stonybrook.edu/`, accessed May 2021), for the normal class the log likelihood is much higher than chance (assumption 1). It is also clear that it is almost always much higher than that of the out-of-class samples (assumption 2), even though the log likelihood for the out-of-class samples is sometimes higher than chance.

We note that where these assumptions do not hold, e.g., for Pima and Vertebral datasets, where the second assumption does not hold, the success rate of our method is relatively low (Tab. 2).

The correlation between the success in detecting anomalies and the classification score is presented in Fig. 5. Presented are both the classification log likelihood (assumption 1) as well as the difference in log likelihood between in-class and out-of-class samples. Evidently, the negative log-likelihood score of assumption 1 is low in almost all cases, but is not highly correlated with the F1 score ($r = -0.1608$, $p = 0.3959$). The difference in classification success that is associated with assumption 2 is, however, highly correlated with the identification performance ($r = -0.7568$, $p = 1.3 \times 10^{-6}$).

We use the same experiments to also check the soft assumption that the original order may be, in some cases, preferable to a random permutation of the features. As can be seen, there is no clear advantage to the original order of the features. In the Mulcross dataset, for example, the permuted variation leads to a stronger distinction between the normal and the anomalous class. The lack of a clear advantage for the original order is further supported by the experimental results in Sec. 4, which show that in 50% of the datasets the original order is preferable and in the rest the permuted order is.

Finally, we explore the counterexample presented in the limitations part of Sec. 5, using the specific parameters for which the normal and the out-of-class samples cannot be distinguished. As can be seen in Fig. 4 (bottom right), in this case the first assumption holds, but the second one does not. This leads to an inability to identify abnormal samples.

## B  ODDS DATASET STATISTICS

Tab. 5 presents the number of samples, the dimensionality, and the number of samples not from the main class for the various datasets that belong to the collection "Multi-dimensional point datasets" from the Outlier Detection DataSets (ODDS, `http://odds.cs.stonybrook.edu/`, accessed May 2021).

## C  ADDITIONAL RESULTS

In Tab. 6 we present results for the second set of experiments, this time reporting AUC (the manuscript reported the F1 scores in Tab. 2.).

In Fig. 7, we present the results of a parameter sensitivity study performed on four datasets. The results indicate that the method is largely insensitive to the parameters: the sliding window size $k$, the dimensionality of the latent space $u$, the softmax temperature $\tau$, and the number of random permutations $r$. In each experiment, the tested parameter was varied, while all other values were default.

In Fig. 8 we show the results for varying $k$ in the range $[1...150]$ for the Arrhythmia dataset. For each $k$ value, the reported results are averages over 10 runs. The best F1 score was obtained at 63 (the $k$ value selected at our default settings for this dataset is 124), but all values were relatively similar and between 58.59% and 67.2% (the AUC values were in the range 80.7%–84.3%). For all $k$, the F1 and AUC are better than all baselines.

In Tab. 7 we show the results for a sensitivity test, in which we set the hyperparameter $k$ as 1 for all datasets, compared to the original results reported in the paper. In 30% of the experiments, this leads to better performance than the $k$ used according to the default setting. The mean F1 is lower by 2.6% (AUC lower by 1.7%), still outperforming all baselines.

Fig. 9 shows the results for the random data experiment, where features are sampled independently from a Gaussian distribution with a different mean per feature, sampled from U[0,1]. These runs are compared to results where the features are sampled from U[-1,1] i.i.d. Also, for each distribution we

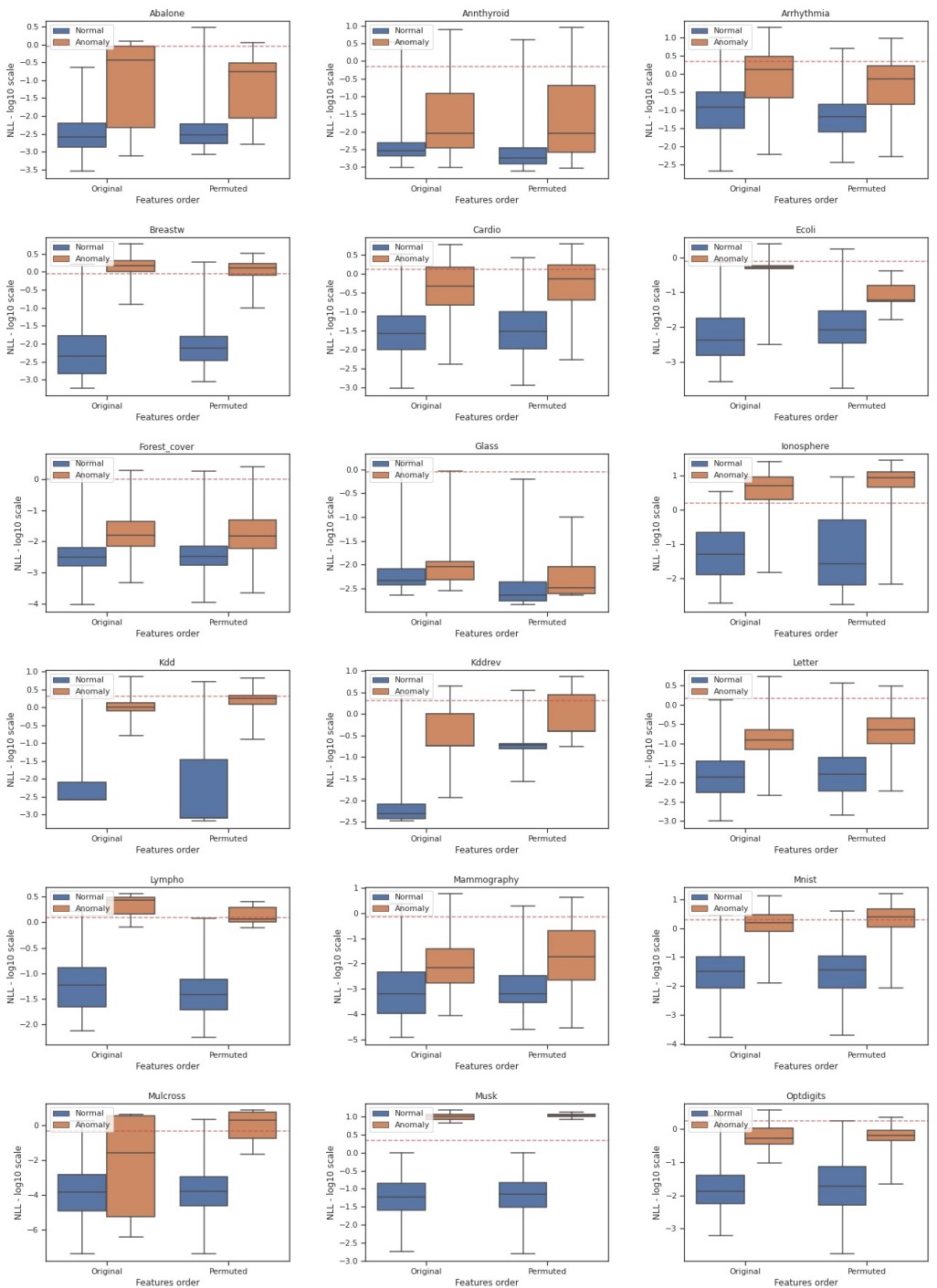

Figure 3: The negative log likelihood (Eq. 1) for matching between the sub-vectors $(a_i^j, b_i^j)$ with networks $F, G$ for multiple ODDS datasets. The results are shown in each plot, from left to right, for in-class samples using the original order of vector elements, out-of-class samples using the same order, in-class samples with a fixed-permutation order of features, out-of-class samples with this permuted order. The samples in the box plots were collected across all test samples $x_i$ and segment indices $j$, without summing over $j$. The horizontal line indicates the negative log likelihood of a random guess. (Part 1/2, continued on the next page)

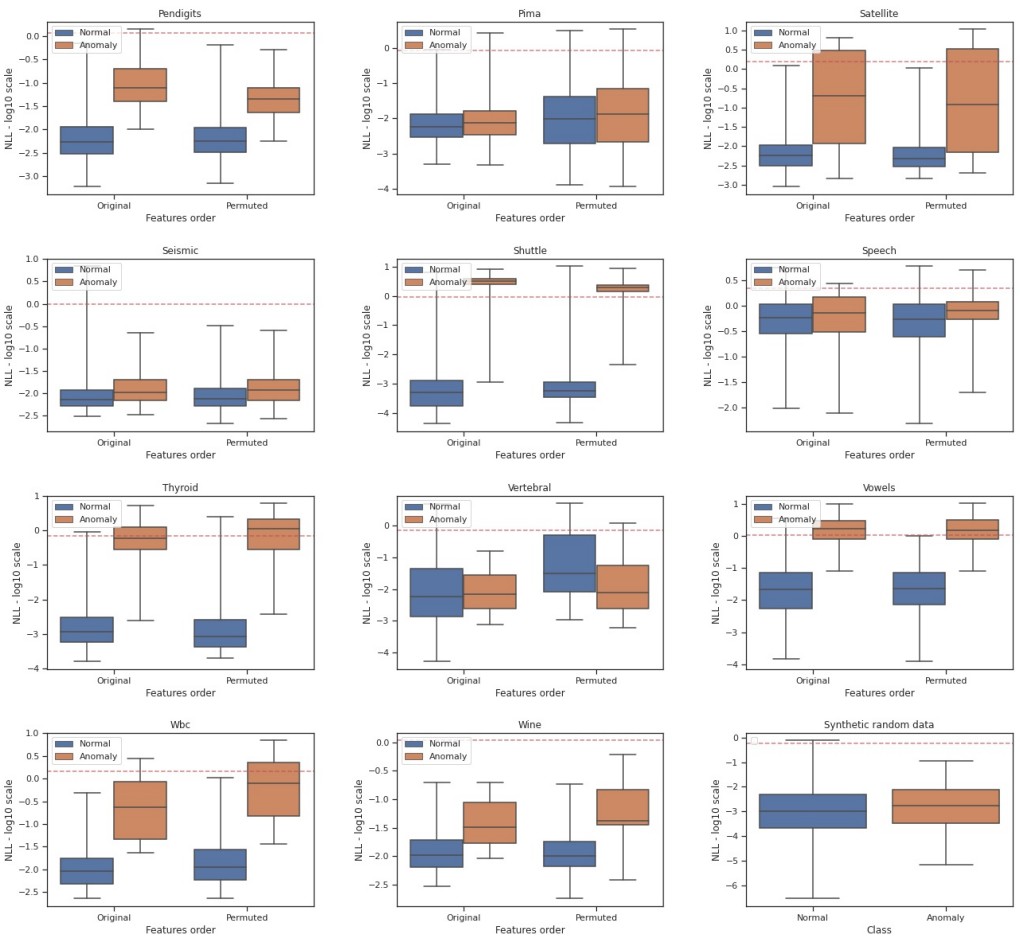

Figure 4: (Part 2/2; see Fig. 3 for details). Bottom right: results on a synthetic dataset that was created to demonstrate a case, for specific $k, r$, in which learning is possible but not in a class dependent manner.

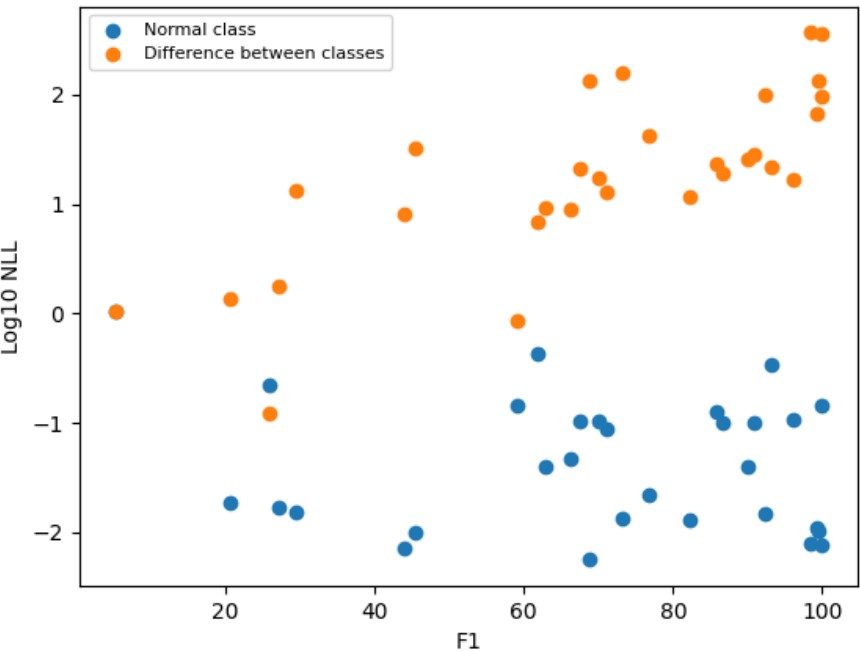

Figure 5: The anomaly detection performance (F1 score, x-axis) vs. the classification performance on the normal class(log likelihood, assumption 1) and the difference in performance between the normal and the anomaly class (assumption 2).

show the results for two variations of our network - one with the default architecture, and one where we change the F network so it is similar to G (i.e. without TanH activation).

As can be seen, for data sampled from Gaussian distribution the network converges considerably faster, which suggests that solving the class-independent task is easier when the features have distinctive differences. Also, in both variations of the data distribution, the TanH architecture variant requires more epochs to converge to similar values, supporting our hypothesis that the design of our architecture makes it harder for the network to learn the class-independent solution.

In Tab. 8 we report the results for running different variations of our architectures: a variant with a single hidden layer for both F and G, one with an extra layer for both, and the default architecture.

In Tab. 9 we report results for a test designed for studying whether the order of the features effects the results of the model. To this end,we measured the performance after employing a single fixed permutation on the datasets, compared to not employing any permutation at all (the original order as provided by the dataset). Permuting the features had a minor effect, and across the collection of datasets, there is no clear advantage to the original order.

Tab. 10 reports the results of the experiment in which both the normal and the out-of-class distributions are sampled from 4D Gaussians with a mean vector of $[1, 2, 3, 4]^\top$ and a per-variable variance of one. The in-class samples where sampled from a convariance matrix with a correlation of 0.85 between variables two and three, but uncorrelated otherwise. For the out-of-class samples, the covariance matrix was diagonal. This is a challenging problem, since variables two and three can appear to be similar even for an out-of-class sample. As can be seen, our method greatly outperforms the baseline methods.

The KNN baseline, while being the simplest baseline, was shown to be the strongest one. In the experiments of the paper we report results for the highest performing parameter, which is $K = 5$ neighbours. The mean F1 score for various values of $K$ tested are: 67.1, 66.3, 67.7, 66.8, 66.4 for $K$ of 1,2,5,10,20, respectively.

## D  ABLATIONS AND PARAMETER SENSITIVITY

We selected four representative datasets ('Wine', 'Glass', 'Thyroid, 'Letter'), which vary in the number of dimensions, the number of samples, and the performance level, and ran an ablation analysis on them. The variants we compare include: (i) a variant of our method, in which the $tanh$ activation of the first layer of $F$ is replaced by a LeakyRELU, (ii) a variant in which only the first of the two normalizations of the query and vectors that $F$ and $G$ output takes place (iii) a variant in which only the second normalization takes place, i.e, normalization occurs in the conventional way, (iv) a variant in which the order of the two normalizations is reversed, and (v) a variant with no normalization, i.e., $F, G$ are trained and used instead of $F^N$ and $G^N$.

The results are reported in Tab. 4. As can be seen, the $tanh$ activation in $F$ improves results, to a varying degree, on the four datasets. Normalization tends to help across datasets. However, on Thyroid, applying no normalization at all provides better results. Omitting only the 2nd normalization, which is often used in contrastive learning, is detrimental in all cases. Omitting the 1st normalization, while keeping the 2nd, also hurts performance. Reversing the order of the two normalizations, which changes the scales of the features before this scale is normalized for, is detrimental.

On the same four datasets we also evaluate the sensitivity of the method to its hyperparameters: $k$, $u$, $\tau$, and $r$. In each experiment, we fix three parameters and vary the fourth. The results, provided in the supplementary Fig. 7 of appendix C, indicate that the method is largely insensitive to its parameters. This robustness is further supported by using the same hyperparameters on a large number of datasets.

To further verify that the method works for any $k$, we tested for Arrhythmia, which has a dimensionality of 274, values of $k$ between 1 and 150. The best F1 score was obtained at 63 (in the experiments it is 124), but all values were relatively similar and between 58.59% and 67.2% (the AUC values were in the range 80.7%–84.3%). For all $k$, the F1 and AUC are better than all baselines, see Appendix C for the full results. Further evidence of the robustness of the $k$ parameter, which is the main hyperparameter, since we fix $u = 200$ and $\tau = 0.01$, is provided by simply using $k = 1$ across all ODDS datasets. In 30% of the experiments, this leads to better performance than the $k$

Table 4: Ablation results (F1 in percents and standard deviation over multiple resampling).

| Variant | Wine | | Glass | | Thyroid | | Letter | |
|---|---|---|---|---|---|---|---|---|
| | $F_1$ Score | SD | $F_1$ Score | SD | $F_1$ Score | SD | $F_1$ Score | SD |
| Ours | **90.0** | 6.3 | **28.9** | 10.6 | 76.8 | 1.2 | **62.8** | 2.4 |
| (i) No $tanh$ | 70.0 | 7.7 | 15.6 | 5.4 | 76.1 | 3.8 | 54.4 | 3.0 |
| (ii) Only 1st norm | 85.0 | 9.2 | 21.1 | 0.07 | 68.7 | 0.02 | 52.2 | 4.7 |
| (iii) Only 2nd norm | 87.0 | 7.8 | 25.6 | 10.0 | 76.5 | 1.8 | 57.2 | 6.6 |
| (iv) 2nd norm then 1st | 84.0 | 4.8 | 22.2 | 11.0 | 68.4 | 2.2 | 57.1 | 3.0 |
| (v) No normalization | 87.0 | 6.4 | 21.1 | 10.5 | **78.1** | 1.8 | 51.0 | 7.2 |

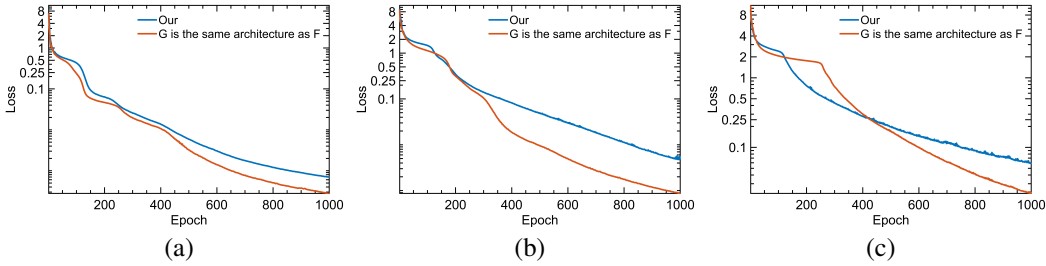

(a)          (b)          (c)

Figure 6: Convergence of the loss on completely random data, in order to evaluate the ability to perform class-independent learning. (a) $d = 6$. (b) $d = 15$. (c) $d = 30$.

used according to Sec. 3. The mean F1 is lower by 2.6% (AUC lower by 1.7%), still outperforming all baselines, see Appendix C for the full results. Note that in this case $k << u$.

We further investigate the sensitivity of our method to the network architecture. When changing both $F$ and $G$, a variant with only one hidden layer obtained, on average on ODDS, an F1 score that is lower by 2.8% (AUC by 1%; still much better than all baselines in both scores), and a variant with an extra hidden layer in both had a mean F1 that is lower by 10.2% (AUC difference of 4%) than the default architecture. See Appendix C for the full tables and Appendix H for a discussion.

As described in Sec. 3, the score is computed multiple times after permuting the features, and the number of repeats ($r$) depends on the dimensionality $d$ and number of samples $n$. In Fig. 7(d), we present the effect of the number of repeats on performance for multiple relatively small datasets, at different performance levels. Shown are the mean F1 over 10 runs and also, as error bars, the SD. We observe that adding repeats typically helps, albeit modestly. It also tends to reduce the variance between runs. A drop in performance between the first and the second repeats may indicate that the order of features is informative. However, this does not happen often.

To study the potential benefit that our method obtains from the particular order of the features in which the data is given, we have compared our results (with no permutations) to those obtained when employing a single, fixed permutation. The average F1 score our method obtains across the entire ODDS collection is 64.1%, and after permutation 64.4% (mean over all datasets, with 10 repeats). This trend, however, is dominated by a few datasets, and in AUC terms, the results are 87.2% vs. 87.1%, where the original order leads. In both AUC and F1 the original order leads in the same 50% of the datasets. The lack of a clear advantage for the original order is also apparent in the auxiliary results of Appendix A.

The same experiment also verifies that our improved performance does not arise solely from the bagging effect of having $r > 1$. Comparing the case of $r = 1$ to the baselines reveals that our average F1 score of 64.1% (AUC 87.2%) is higher by a sizable margin then all of GOAD's architectures, which have a mean F1 of 51.3–55.9% (depending on the architecture; AUC of 79.6–81.0%) and COPOD's 44.7% mean F1 score (AUC 79.1%).

# E    LIMITATIONS

In some symmetric cases, for specific parameters and with no permutation, our method would fail in the task of anomaly detection. Specifically, any scenario in which for both normal and anomalous data the pairs constructed from each instance $\{(a_i^j, b_i^j)\}_{j=1}^m$ are identical would be impossible for the network to differentiate between. For example normal data of the form (0,1,0,0,0) and anomalous data (0,0,0,1,0) with $k = 2$ and $r = 1$ violates the second assumption listed in Sec. 1. See Appendix A for simulations of this example.

# F    AN ALTERNATIVE METHOD (REPORTING A NEGATIVE RESULT)

We briefly summarize an alternative method that was tried, which failed to produce the results we hoped for. Earlier in the development process, we had an intuition that making a comparison with external samples might be beneficial for our model, and we experimented with drawing the negative samples from other instances, i.e. instead of using:

$$\boldsymbol{q} = F(b_i^j), \boldsymbol{v}^+ = G(a_i^j), \boldsymbol{v}^- = G(a_i^{j'}) \tag{2}$$

for some $i, j, j' \neq j$, we used:

$$\boldsymbol{q} = F(b_i^j), \boldsymbol{v}^+ = G(a_i^j), \boldsymbol{v}^- = G(a_{i'}^j) \tag{3}$$

for $i, i' \neq i, j$. The results obtained by employing this method on the ODDS repository were significantly lower in terms of F1 and AUC performance on almost all datasets, and were not competitive with the baseline methods.

# G    RUNTIME

In Tab. 11 we show the average runtime in seconds for each of the datasets. We include two versions: one with the default amount of repeats $r_{default} = 1 + \lfloor 100(\log(n) + d)^{-1} \rfloor$ and another one with $r_{faster} = min(2, r_{default})$.

We observe that the difference in performance between the two versions is typically minor, and the faster method might be preferred if runtime is a top priority. The code includes an option for running the model in the faster mode.

The google colab infrastructure was used to run the experiments. GPU: Tesla K80, 12GB GDDR5 VRAM; CPU: Single core Xeon Processors @2.3Ghz; RAM: 24 GB. For the GOAD baseline that required larger memory, we used a 32GB GPU and 512GB RAM.

# H    DISCUSSION OF ALTERNATIVE ARCHITECTURES

The architecture we employ for F and G is a straightforward fully connected architecture with two hidden layers. One can hypothesize that with less capacity, the network may not be able to learn the required mappings as effectively, making it incapable of utilizing the first assumption. However, with added capacity, the network may shift from the class specific solution to a generic solution, validating our 2nd assumption. See Galanti et al. (2018) for a remotely related phenomenon in the completely different unsupervised learning task of mapping between two visual domains: with too few layers the mapping fails to produce samples in the target domain, for too many layers, the mappings are not specific to the input image.

The hypothesis is partly validated empirically in the results reported in Appendix. C Tab. 8. A variant with only one hidden layer obtained only 1% less mean AUC (still better by a gap than all baselines) on the ODDS benchmark than our original results, and a variant with an extra hidden layer had an AUC 4% lower.

In order to study whether this drop in performance is due to assumption 2, we examine the average log likelihood. The mean loss for the "normals" category is 0.016 and 0.014 for the original architechture

and one extra layer respectively, and the mean anomaly class losses are 1.54 and 0.96. We, therefore, observe that the ability to solve the problem (assumption 1) is almost the same and slightly improves with the added capacity. However, the gap between the normal class and the outlier samples has shrunk dramatically with the extra layer.

Table 5: The number of samples, the dimensionality, and the number of samples not from the main class for the datasets used in the second set of experiments.

| Dataset | n | $d$ | Outliers |
|---|---|---|---|
| Wine | 129 | 13 | 10 (7.7%) |
| Lympho | 148 | 18 | 6 (4.1%) |
| Glass | 214 | 9 | 9 (4.2%) |
| Vertebral | 240 | 6 | 30 (12.5%) |
| WBC | 278 | 30 | 21 (5.6%) |
| Ecoli | 336 | 7 | 9 (2.6%) |
| Ionosphere | 351 | 33 | 126 (36%) |
| Arrhythmia | 452 | 274 | 66 (15%) |
| BreastW | 683 | 9 | 239 (35%) |
| Pima | 768 | 8 | 268 (35%) |
| Vowels | 1456 | 12 | 50 (3.4%) |
| Letter Recognition | 1600 | 32 | 100 (6.25%) |
| Cardio | 1831 | 21 | 176 (9.6%) |
| Seismic | 2584 | 11 | 170 (6.5%) |
| Musk | 3062 | 166 | 97 (3.2%) |
| Speech | 3686 | 400 | 61 (1.65%) |
| Thyroid | 3772 | 6 | 93 (2.5%) |
| Abalone | 4177 | 9 | 29 (0.69%) |
| Optdigits | 5216 | 64 | 150 (3%) |
| Satimage-2 | 5803 | 36 | 71 (1.2%) |
| Satellite | 6435 | 36 | 2036 (32%) |
| Pendigits | 6870 | 16 | 156 (2.27%) |
| Annthyroid | 7200 | 6 | 534 (7.42%) |
| Mnist | 7603 | 100 | 700 (9.2%) |
| Mammography | 11183 | 6 | 260 (2.32%) |
| Shuttle | 49097 | 9 | 3511 (7%) |
| KDDCUP99-Rev | 121597 | 120 | 24319 (20%) |
| Mulcross | 262144 | 4 | 26214 (10%) |
| ForestCover | 286048 | 10 | 2747 (0.9%) |
| KDDCUP99 | 494021 | 120 | 97277 (19.6%) |

Table 6: AUC results for the ODDS benchmarks. In order to avoid selecting the architecture for the baseline methods, we used all available versions of these methods. The missing DROCC experiments are due to the limitations of the published code.

| Method | DROCC (Thyroid) | DROCC (Arrhythmia) | DROCC (Abalone) | GOAD (Thyroid) | GOAD (kddrev) | GOAD (kdd) | COPOD | Ours |
|---|---|---|---|---|---|---|---|---|
| Wine | 53.5±22.3 | 60.1±32.3 | 90.9±8.2 | 95.2±1.9 | 97.3±1.7 | 86.3±9.5 | 87.5±1.7 | **99.5**±0.6 |
| Lympho | 6.4±5.2 | 58.6±30.4 | 83.7±12.4 | 94.8±5.6 | 79.7±11.1 | 59.9±14.9 | 99.4±0.4 | **99.5**±0.3 |
| Glass | 63.5±9.1 | 55.5±21.8 | 75.4±8.9 | 62.2±14.0 | 85.5±7.0 | 82.1±6.3 | 63.7±3.3 | **88.1**±5.0 |
| Vertebral | 55.0±5.1 | **58.0**±15.4 | 41.2±10.1 | 47.0±12.8 | 52.2±3.9 | 49.4±4.2 | 32.6±1.2 | 51.1±3.2 |
| WBC | 6.8±1.8 | 41.3±25.0 | 35.4±13.1 | 95.4±0.7 | 66.1±11.5 | 86.6±2.9 | **96.3**±0.5 | 95.4±1.1 |
| Ecoli | N/A | N/A | N/A | 82.7±8.4 | **87.2**±3.3 | 84.7±6.8 | 81.0±1.2 | 86.5±1.2 |
| Ionosphere | 19.6±5.8 | 83.5±5.6 | 80.0±2.8 | 92.4±1.3 | 96.3±1.1 | 96.5±1.1 | 80.3±2.1 | **98.1**±0.4 |
| Arrhythmia | 53.2±7.0 | 52.7±8.6 | 51.2±8.1 | 80.0±1.9 | 73.3±5.1 | 64.3±8.8 | 80.5±1.3 | **81.7**±0.6 |
| Breastw | 7.7±8.6 | 64.4±33.0 | 96.6±3.3 | 98.7±0.8 | 80.8±9.5 | 97.7±0.8 | **99.4**±0.2 | 99.1±0.3 |
| Pima | 36.2±4.6 | 54.9±11.0 | **69.1**±4.9 | 68.7±3.9 | 59.3±2.2 | 63.2±2.3 | 65.2±0.7 | 59.4±2.8 |
| Vowels | 79.4±9.5 | 72.0±11.9 | 95.3±2.1 | 81.0±2.4 | 98.5±0.3 | 97.6±0.5 | 49.6±1.0 | **99.7**±0.1 |
| Letter | 77.6±3.3 | 73.3±5.4 | 90.0±1.2 | 60.9±0.7 | 89.9±0.5 | 87.6±0.9 | 50.1±0.8 | **92.8**±0.9 |
| Cardio | 84.3±4.0 | 73.8±11.8 | 73.5±3.2 | **94.8**±1.7 | 81.3±4.5 | 84.6±3.0 | 92.2±0.3 | 92.7±0.8 |
| Seismic | 58.2±2.8 | 60.3±4.5 | 56.7±1.3 | 69.5±1.5 | 67.2±1.2 | 67.9±1.2 | **70.8**±0.4 | 62.9±1.0 |
| Musk | 2.3±5.1 | **100.0**±0.0 | **100.0**±0.0 | **100.0**±0.0 | **100.0**±0.0 | **100.0**±0.0 | 94.5±0.2 | **100.0**±0.0 |
| Speech | 51.2±5.6 | 50.5±4.0 | 52.6±3.4 | 47.1±1.3 | **65.3**±3.2 | 54.1±4.4 | 49.1±0.5 | 58.9±2.7 |
| Thyroid | 95.6±0.9 | 96.1±2.5 | 98.1±0.3 | 94.5±1.5 | 77.1±8.8 | 89.2±3.0 | 94.1±0.2 | **98.5**±0.1 |
| Abalone | 82.4±13.8 | 52.9±25.8 | 70.6±9.7 | 89.2±0.9 | 46.0±3.7 | 54.3±7.8 | 86.3±0.3 | **94.3**±0.6 |
| Optidigits | 84.2±4.6 | 89.0±4.6 | 89.5±2.1 | 66.9±3.3 | 95.7±0.5 | 93.1±1.9 | 68.0±0.4 | **97.5**±1.5 |
| Satimage | 19.1±1.4 | 87.5±8.8 | 11.5±1.2 | 99.1±0.1 | 86.5±7.1 | 93.2±1.7 | 97.4±0.1 | **99.8**±0.1 |
| Satellite | 64.6±8.9 | 73.1±1.3 | 50.2±2.2 | 69.1±0.8 | 76.3±1.0 | 78.2±0.9 | 63.5±0.2 | **80.6**±1.7 |
| Pendigits | 58.9±7.6 | 50.8±15.4 | 76.6±5.4 | 87.5±3.9 | 89.2±2.9 | 85.1±3.4 | 90.4±0.2 | **99.5**±0.1 |
| Annthyroid | 92.9±2.3 | 86.5±3.6 | 93.4±1.3 | 76.1±6.5 | 89.6±4.9 | **93.2**±0.9 | 77.4±0.4 | 80.5±1.3 |
| MNIST | N/A | N/A | N/A | 90.9±0.4 | 89.4±0.7 | 87.7±1.0 | 77.2±0.2 | **98.2**±0.0 |
| Mammography | 81.0±1.3 | 85.0±2.1 | 82.0±1.5 | 66.3±6.4 | 57.2±1.9 | 54.5±2.3 | **90.5**±0.1 | 81.1±2.0 |
| Mulcross | N/A | N/A | N/A | 100.0±0.0 | N/A | 51.3±15.8 | 93.2±0.0 | **100.0**±0.0 |
| Shuttle | N/A | N/A | N/A | 88.4±5.5 | N/A | 99.9±0.0 | 99.4±0.0 | **100.0**±0.0 |
| Forest | N/A | N/A | N/A | 15.9±6.6 | N/A | 76.0±5.3 | 88.4±0.0 | **96.2**±0.6 |
| KDD | N/A | N/A | N/A | 92.8±1.0 | N/A | 99.8±0.0 | 78.6±0.1 | **99.9**±0.0 |
| KDD-Rev | N/A | N/A | N/A | N/A | N/A | 99.7±0.2 | 75.1±0.2 | **99.8**±0.1 |

Table 7: F1 and AUC results for $k = 1$ and the default settings of $k$ mentioned in the paper

| | F1 | | AUC | |
|---|---|---|---|---|
| Method | $k = 1$ | $k = default$ | $k = 1$ | $k = default$ |
| Wine | 88.0 | 90.0 | 99.3 | 99.5 |
| Lympho | 86.7 | 86.7 | 99.5 | 99.5 |
| Glass | 22.2 | 27.2 | 87.3 | 88.1 |
| Vertebral | 28.0 | 26.0 | 52.4 | 51.1 |
| WBC | 71.4 | 67.6 | 95.6 | 95.4 |
| Ecoli | 62.2 | 70.0 | 86.0 | 86.5 |
| Ionosphere | 93.2 | 93.2 | 97.8 | 98.1 |
| Arrhythmia | 59.7 | 61.8 | 81.0 | 81.7 |
| Breastw | 94.8 | 96.1 | 98.4 | 99.1 |
| Pima | 60.0 | 59.1 | 60.5 | 59.4 |
| Vowels | 86.4 | 90.8 | 99.5 | 99.7 |
| Letter | 56.8 | 62.8 | 89.5 | 92.8 |
| Cardio | 72.0 | 71.0 | 92.1 | 92.7 |
| Seismic | 23.4 | 20.7 | 64.9 | 62.9 |
| Musk | 100.0 | 100.0 | 100.0 | 100.0 |
| Speech | 3.6 | 5.2 | 54.2 | 58.9 |
| Thyroid | 78.1 | 76.8 | 98.1 | 98.5 |
| Abalone | 69.7 | 68.7 | 93.1 | 94.3 |
| Optidigits | 37.9 | 66.3 | 92.3 | 97.5 |
| Satimage | 91.8 | 92.4 | 99.7 | 99.8 |
| Satellite | 72.6 | 73.2 | 81.1 | 80.6 |
| Pendigits | 71.0 | 82.3 | 98.7 | 99.5 |
| Annthyroid | 45.5 | 45.4 | 79.6 | 80.5 |
| MNIST | 82.0 | 85.9 | 96.4 | 98.2 |
| Mammography | 28.9 | 29.4 | 81.5 | 81.1 |
| Shuttle | 98.3 | 98.4 | 100.0 | 100.0 |
| KDD-Rev | 99.0 | 99.2 | 99.8 | 99.8 |
| Mulcross | 100.0 | 100.0 | 100.0 | 100.0 |
| Forest | 28.3 | 44.0 | 86.6 | 96.2 |
| KDD | 98.6 | 99.4 | 99.9 | 99.9 |
| mean | 67.0 | 69.6 | 88.8 | 89.7 |

Table 8: F1 and AUC results for variations of the architecture: Using only one hidden layer, adding an extra hidden layer and the default architecture

| Method | F1 | | | AUC | | |
|---|---|---|---|---|---|---|
| | Single-layer | Extra-layer | Default | Single-layer | Extra-layer | Default |
| Wine | 74.0 | 88.0 | 90.0 | 96.8 | 98.3 | 99.5 |
| Lympho | 86.7 | 76.7 | 86.7 | 99.5 | 98.6 | 99.5 |
| Glass | 17.8 | 13.3 | 27.2 | 84.2 | 82.0 | 88.1 |
| Vertebral | 16.7 | 30.0 | 26.0 | 48.3 | 59.1 | 51.1 |
| WBC | 69.5 | 67.6 | 67.6 | 94.8 | 94.4 | 95.4 |
| Ecoli | 57.8 | 62.2 | 70.0 | 86.6 | 87.4 | 86.5 |
| Ionosphere | 94.4 | 91.6 | 93.2 | 98.3 | 97.3 | 98.1 |
| Arrhythmia | 62.1 | 62.7 | 61.8 | 80.9 | 81.5 | 81.7 |
| Breastw | 95.4 | 95.6 | 96.1 | 98.6 | 98.6 | 99.1 |
| Pima | 59.5 | 57.1 | 59.1 | 60.1 | 57.2 | 59.4 |
| Vowels | 90.0 | 73.6 | 90.8 | 99.6 | 97.8 | 99.7 |
| Letter | 51.2 | 28.2 | 62.8 | 87.0 | 70.1 | 92.8 |
| Cardio | 66.6 | 78.4 | 71.0 | 90.3 | 96.5 | 92.7 |
| Seismic | 23.2 | 19.8 | 20.7 | 61.0 | 63.3 | 62.9 |
| Musk | 100.0 | 67.0 | 100.0 | 100.0 | 96.8 | 100.0 |
| Speech | 4.9 | 2.3 | 5.2 | 53.3 | 45.2 | 58.9 |
| Thyroid | 73.3 | 58.1 | 76.8 | 97.5 | 96.8 | 98.5 |
| Abalone | 68.3 | 56.6 | 68.7 | 94.2 | 92.6 | 94.3 |
| Optidigits | 47.1 | 2.7 | 66.3 | 94.0 | 72.0 | 97.5 |
| Satimage | 91.0 | 91.0 | 92.4 | 99.7 | 99.7 | 99.8 |
| Satellite | 73.3 | 69.2 | 73.2 | 79.5 | 75.8 | 80.6 |
| Pendigits | 85.1 | 54.6 | 82.3 | 99.6 | 94.2 | 99.5 |
| Annthyroid | 43.7 | 43.6 | 45.4 | 79.7 | 78.0 | 80.5 |
| MNIST | 82.3 | 52.4 | 85.9 | 96.7 | 86.1 | 98.2 |
| Mammography | 31.7 | 27.7 | 29.4 | 81.1 | 68.4 | 81.1 |
| Shuttle | 98.0 | 98.1 | 98.4 | 99.9 | 100.0 | 100.0 |
| KDD-Rev | 95.0 | 96.5 | 99.2 | 98.4 | 98.9 | 99.8 |
| Mulcross | 100.0 | 94.1 | 100.0 | 100.0 | 99.3 | 100.0 |
| Forest | 49.2 | 13.6 | 44.0 | 97.0 | 83.5 | 96.2 |
| KDD | 97.4 | 99.1 | 99.4 | 99.8 | 100.0 | 99.9 |
| mean | 66.8 | 59.0 | 69.6 | 88.6 | 85.6 | 89.7 |

Table 9: F1 and AUC results for: no permutations at all ($r = 0$) and for one fixed permutation ($r = 1$)

|              | F1 | | AUC | |
| --- | --- | --- | --- | --- |
| Method | $r = 0$ | $r = 1$ | $r = 0$ | $r = 1$ |
| Wine | 74.0 | 73.2 | 95.1 | 94.7 |
| Lympho | 85.0 | 82.8 | 99.4 | 99.1 |
| Glass | 12.2 | 22.9 | 74.5 | 81.5 |
| Vertebral | 14.3 | 14.2 | 52.6 | 42.3 |
| WBC | 63.3 | 65.3 | 94.1 | 93.5 |
| Ecoli | 66.7 | 57.8 | 87.6 | 86.4 |
| Ionosphere | 90.6 | 92.8 | 96.5 | 97.8 |
| Arrhythmia | 62.4 | 61.3 | 80.4 | 81.0 |
| Breastw | 95.1 | 94.5 | 98.5 | 98.1 |
| Pima | 59.3 | 58.5 | 59.2 | 58.8 |
| Vowels | 80.4 | 79.9 | 98.4 | 98.4 |
| Letter | 49.6 | 55.3 | 88.7 | 89.2 |
| Cardio | 66.8 | 67.0 | 89.7 | 89.8 |
| Seismic | 20.8 | 19.8 | 62.6 | 60.1 |
| Musk | 100.0 | 100.0 | 100.0 | 100.0 |
| Speech | 3.6 | 2.0 | 52.3 | 55.5 |
| Thyroid | 75.5 | 69.8 | 98.3 | 96.4 |
| Abalone | 61.4 | 64.1 | 88.5 | 92.2 |
| Optidigits | 68.5 | 55.8 | 97.6 | 95.9 |
| Satimage | 90.6 | 91.7 | 99.7 | 99.8 |
| Satellite | 73.8 | 73.3 | 83.3 | 80.6 |
| Pendigits | 69.8 | 69.8 | 97.6 | 97.7 |
| Annthyroid | 41.9 | 44.6 | 72.9 | 79.8 |
| MNIST | 83.7 | 84.8 | 97.3 | 97.9 |
| Mammography | 21.9 | 24.8 | 75.9 | 73.7 |
| Shuttle | 97.7 | 98.0 | 99.9 | 99.9 |
| KDD-Rev | 99.2 | 99.2 | 99.9 | 99.8 |
| Mulcross | 73.5 | 75.8 | 90.5 | 86.7 |
| Forest | 23.5 | 35.1 | 86.2 | 89.5 |
| KDD | 98.9 | 98.3 | 99.9 | 99.9 |
| mean | 64.1 | 64.4 | 87.2 | 87.2 |

Table 10: AUC results for a synthetic data experiment with 4D data that has the same class mean for both classes, only that in the normal class variables two and three are correlated. Shown are mean results and Standard Deviations over 10 repeats.

| Method | AUC (percent; Mean $\pm$ SD) |
| --- | --- |
| COPOD (Li et al., 2020) | $52.2 \pm 3.2$ |
| Isolation Forest (Liu et al., 2008) | $62.3 \pm 4.6$ |
| OC-SVM (Schölkopf et al., 1999) | $66.8 \pm 1.1$ |
| GOAD KDD architecture (Bergman & Hoshen, 2020) | $74.0 \pm 3.2$ |
| GOAD Thyroid architecture (Bergman & Hoshen, 2020) | $73.7 \pm 1.3$ |
| Ours | $88.1 \pm 1.5$ |

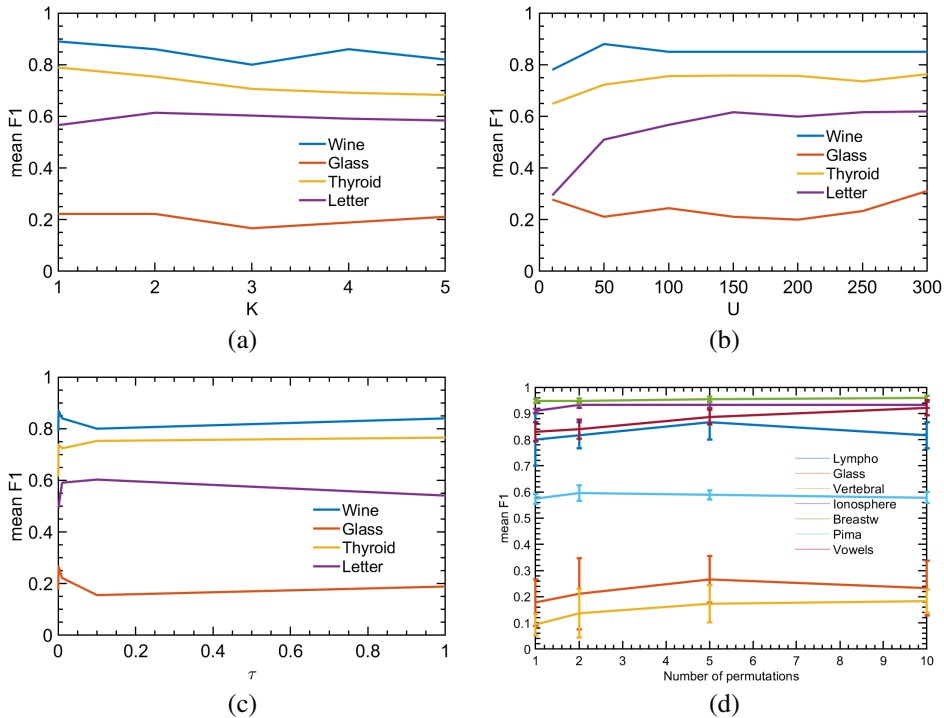

Figure 7: Sensitivity with respect to (a) $k$, (b) $u$, (c) $\tau$, and (d) the number of random permutations $r$. All other values are taken at the default value.

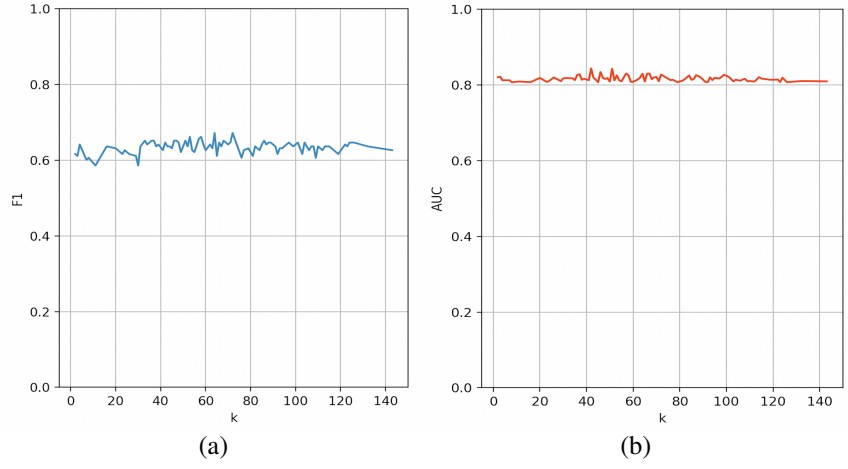

Figure 8: Varying k in the Arrhythmia dataset. (a) F1 results (b) AUC results.

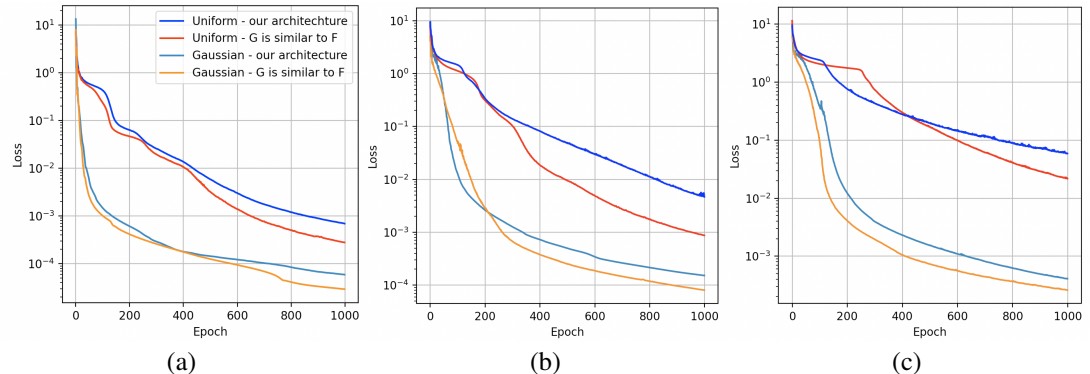

(a)             (b)             (c)

Figure 9: Convergence of the loss on random data, sampled from uniform and Gaussian distributions. (a) $d = 6$. (b) $d = 15$. (c) $d = 30$.

Table 11: Runtime in seconds and F1-Score for $r_{default} = 1 + \lfloor 100(\log(n) + d)^{-1} \rfloor$ and $r_{faster} = min(2, r_{default})$.

| Dataset | $r_{default}$ runtime | $r_{faster}$ runtime | $r_{default}$ F1 | $r_{faster}$ F1 |
|---|---|---|---|---|
| wine | 21.1 | 8.0 | 90.0% | 73.0% |
| lympho | 35.5 | 15.0 | 86.7% | 83.3% |
| glass | 24.9 | 5.2 | 27.2% | 25.6% |
| vertebral | 30.3 | 5.8 | 26.0% | 17.0% |
| wbc | 33.2 | 22.7 | 67.6% | 66.7% |
| ecoli | 35.5 | 8.4 | 70.0% | 66.7% |
| ionosphere | 28.0 | 18.8 | 93.2% | 92.5% |
| arrhythmia | 33.5 | same | 61.8% | same |
| breastw | 38.9 | 12.2 | 96.1% | 94.6% |
| pima | 27.1 | 9.8 | 59.1% | 58.2% |
| vowels | 83.7 | 28.0 | 90.8% | 86.6% |
| abalone | 44.4 | 8.4 | 68.7% | 66.2% |
| letter | 107.6 | 72.5 | 62.8% | 59.5% |
| cardio | 105.9 | 52.8 | 71.0% | 67.8% |
| seismic | 159.9 | 79.2 | 20.7% | 21.0% |
| musk | 381.3 | same | 100.0% | same |
| speech | 662.4 | same | 5.2% | same |
| thyroid | 76.7 | 17.1 | 76.8% | 73.1% |
| optdigits | 332.5 | same | 66.3% | same |
| satimage | 430.4 | 286.8 | 92.4% | 92.1% |
| satellite | 336.7 | 222.3 | 73.2% | 74.7% |
| pendigits | 224.1 | 89.9 | 82.3% | 76.3% |
| annthyroid | 80.2 | 21.7 | 45.4% | 43.5% |
| mnist | 223.1 | same | 85.9% | same |
| mammography | 150.8 | 42.5 | 29.4% | 24.2% |
| shuttle | 212.4 | 89.1 | 98.4% | same |
| forest_cover | 240.3 | 107.9 | 44.0% | 37.5% |
| mullcross | 279.5 | 43.6 | 100.0% | 99.0% |
| kdd | 1674.1 | same | 99.4% | same |
| kddrev | 1771.3 | same | 99.2% | same |

