# OpenReview forum: "Anomaly Detection for Tabular Data with Internal Contrastive Learning"
_ICLR.cc/2022/Conference — ICLR 2022 Poster_

### Official Review · Reviewer_e9Pz · 2021-10-29

**Correctness:** 3
**Technical Novelty And Significance:** 2
**Empirical Novelty And Significance:** 3
**Recommendation:** 6
**Confidence:** 4

**Main Review:**

1. Assumption
- In the introduction, the authors assume that the feature relationship is class-dependent.
- It would be good if the authors can provide some examples about this assumption and whether this assumption is practical.

2. Consecutive features & Same number of features
- The authors define the "consecutive features".
- However, in tabular data, the order of the features are independent.
- In that point of view, is there any reason that the authors are utilizing the "consecutive" features to extract the subset of the features?
- Also, can we provide some more flexibility for selecting the subsets of the features? Like a different number of features?

3. Intuitions in Figure 1
- What is the intuition of selecting negative pairs?
- I understand that the positive pair can be a_i^j and b_i^j.
- However, it is hard to understand why the negative pair is a_i^m (where m is not j).
- For instance, in Figure 1, a_i^6 is a subset of b_i^3. I understand that we use different embedding functions. However, still the inputs are the subset.
- Maybe we can set the negative pair as a_m^j where (m is not j).

4. Hyper-parameter optimization
- It seems like in the problem setting, we do not have a validation set. (Because we only have in-sample data).
- In that case, how did the authors optimize the hyper-parameters of the models? Like the encoder dimensions, learning rates, k, etc?
- There are some rules of thumbs in page 4. But not sure how they are determined. Are they determined by checking the testing performances? If yes, I think it is a somewhat unfair way of doing the experiments.
- Especially, it seems very heuristic of the thresholds determined by the dimensions (d). d < 40, d> 160, etc.

5. Baselines
- The authors have relevant baselines in the paper. However, some are missing.
- Maybe it would be good if the authors add the following baselines: (i) Isolated Forests, (ii) PIDForest.

6. Sensitive analyzes
- The authors provide various sensitive analyzes in the appendix and those are very helpful.
- However, in Figure 6 and 7, with very small k (like k = 1), the performances are similar to large k which is somewhat weird for me.
- For instance, if k = 1, it means that we do contrastive learning between one feature and d-1 features which makes highly imbalanced networks F and G.
- I am not sure whether this kind of learning would achieve meaningful representations for anomaly detection.

**Summary Of The Paper:**

- The authors propose a novel anomaly detection framework that can be useful for tabular data.
- The authors utilize the contrastive learning to learn the relationship between features in the tabular data. Then, using the loss as the anomaly scores for evaluation.
- The authors provide extensive experimental results which are promising.

**Summary Of The Review:**

Strength
- The experimental results are quite promising and extensive.
- The proposed method is simple and easy to reproduce.

Weakness
- The intuitions of the proposed method is limited. It is actually my biggest concern on this paper. If this can be improved during the rebuttal period, I will increase my scores. Please see my detailed comments above.
- Some sensitive analyses are somewhat weird (e.g., k = 1).
- Some additional baselines are needed to complete the experiments.

-----------------------------------

Thank you for the detailed responses.
I carefully read the responses.
Some concerns (e.g., assumptions, consecutive features, and additional baselines) are resolved. Thank you.
However, there are still some remaining questions.

1. Intuition for negative pairing.
- It is still not clear why the authors select those pairs as the negative pairs?
- In figure 1, b_i^3 and a_i^6 are highly overlapped but still negative pair.
- It would be good if the authors can explain about this clearer with proper intuitions.

2. k=1
- If k = 1, we do contrastive learning with the feature (with dimension = 1) and the feature (with dimension = d-1).
- If the entire feature dimensions are large, the differences would be larger.
- I am not sure whether this is the right way to do contrastive learning between these large different features.

Although other reviewers are leaning to acceptance, I will stay with my original scores (5) until those above concerns are resolved.

Thank you.

---

> ### Author Response · Authors · 2021-11-16
> **Thank you for the very comprehensive review (part 1 of the reply)**
>
> Thank you for the very comprehensive review, which will no doubt help improve the manuscript.
>
> 1. Assumptions: this assumption is referred to in the paper as “assumption 2” and is validated in Appendix A (Fig. 4,5) by showing the log-likelihood of the prediction for the correct class vs. other classes.
> Following the review, we describe below the probabilities obtained for specific samples in the ``Thyroid’’ dataset for in-class and out-of-class samples, in the case of k=1, when masking the T3 feature. For in-class samples, the network always predicts T3 as the masked feature. However, for out-of-class samples, the network often predicts the feature T4U. This makes sense, since T3 levels vary greatly between in-class and out-of-class samples.
>
> 2. (a.) The method does not rely on using consecutive features and the use of consecutive features is only a matter of implementation. As we show, the results are very similar when features are permuted and other subsets of features are taken. Please see the 2nd paragraph on page 8 (submission version) for the experimental results. Our conclusion, as stated in the paper is that “the original order [i.e., consecutive features] does not provide a clear advantage over a random order of features”. In other words, despite the fact that the order in which the features are given in the dataset may not be random, our method does not benefit from employing this specific order instead of any other random order of features.
> Calling the use of consecutive features “a soft assumption” instead of “a straightforward implementation choice” seems to be confusing, and in the revised version (end of Sec. 1) we add wording to clear this confusion. As noted, this implementation choice does not seem to be preferable to the alternative of selecting random non-overlapping subsets.
> (b.) The option to select $k$ of varying size within the same experiment would require training multiple networks (due to the varying input size), or applying padding. While this is entirely possible, it would reduce the elegance of the method and introduce additional hyperparameters.
>
> 3. On the intuitive level:
> (i) Unlike structured datatypes such as images, audio, and data with known dependencies, the only structure in tabular data arises from the statistical link between different subsets of the elements.
> (ii) This link has been exploited in the past for tasks such as data imputation.
> (iii) If the imputed part differs from the expectation it could mean that the datapoint is an outlier.
> (iv) Integrating this information over many imputation attempts would lead to more accurate detection of outliers.
> (v) Contrastive learning is proving to be a very powerful way to study relations between pairs of views.
> (vi) One can use as the background distribution for obtaining negative samples either other sub-vectors or same location subvectors from other samples. However, the second option is much more difficult since the same set of features can be very similar in some of the other samples from the same class. This is true for both the masked part, and the unmasked part. Either one can be the same in other samples of the same class.
> Consider, for example, the Thyroid dataset. The in-class samples are of patients diagnosed as normal or sub-normal functioning. It is unlikely that one can estimate the level of one thyroid hormone given the rest in an accuray that would allow one to match, in a very high accuracy, between some blood measurements of a patient and the rest.  However, as we discuss in point 1(a) above, finding out, for each class, the specific features that are missing (vs. other subset of features) is entirely feasible.
>
> [continued below]

---

> > ### Author Response · Authors · 2021-11-16
> > **part 2 of the reply**
> >
> > Still addressing the third issue the reviewer raised, below are three relevant points that are slightly more technical:
> >
> > (a.) As for using different samples as the negative part of the contrastive loss -- we report such attempts as the negative results described in appendix D.
> >
> > (b.) The ability to select the correct missing sub-vector, given the complementary part, can be reduced to the ability to [i] recognize the location of the missing features and [ii] perform imputation. Both of these tasks are feasible. Therefore, the self-supervised task is feasible.
> >
> > (c.) As for the vectors being subsets of one another -- we address this as part of the deterministic algorithm we propose. “Note that the contrastive learning task that $F$ and $G$ are trained to solve can be easily solved with a very short deterministic program that is class-independent, which checks the overlap between $b_i^j$ and $a_i^j$. However, the success of our method implies that by training our neural method, one learns class-specific models. Furthermore, simulations on synthetic data demonstrate that learning the class-independent solution requires a lengthy training process and does not converge to a perfect classifier. These results, which shed light on self-supervised training, are given in Sec. 5.”
> >
> > For us, this is an eye-opening phenomenon that teaches us about the nature of self-supervised representation learning. As discussed in Sec. 5, many of the most successful self-supervised methods rely on image transformations, and can be solved by a simple deterministic computer vision algorithm. This program would, given two images, decide if there is a transformation that links them. The fact that despite this simple image-level reduction one can learn very powerful semantic representation by using networks teaches us much about the role of capacity and inductive bias as ways of obtaining meaning. In other words, semantic information can emerge from the data more efficiently than  “simpler” low-level solutions.
> >
> >
> > 4. A strong point in favor of our method is the lack of sensitivity to its parameters. Our method is very stable to the choice of parameters, and together with the very sizable performance gap over the previous methods, we can use an almost arbitrary set of fixed parameters and still show markedly better accuracy than the existing methods.
> > The rule that depends on the dimensionality $d$ stems from the large variance of this value among the datasets and the wish to use smaller $k$ for smaller $d$. However, one can simply use $k=1$ and still outperform the existing methods by a large margin, as the reviewer noted.
> >
> > 5. Following the review, we have added these baselines. Two other baselines were added to the list of long baselines in the manuscript at the request of reviewer iWxT. None of the new baselines outperfoms our method.
> >
> > 6. We provide our code to ensure reproducibility and have verified this result. While encoding a scalar is unintuitive (although see, e.g., the low-dimensional encoding that takes place in positional encoding), it is ultimately an elaborate imputation operation. The main differences from conventional imputation are that (1) a contrastive loss is used, and (2) the index of the missing element is unknown. Since (2) can be reliably inferred from the observed elements, this seems entirely plausible.

---

> ### Author Response · Authors · 2021-11-26
> **Response to the modified review**
>
> We are sorry to have missed the update that took place on Nov 24. The openreview system notifies us of new comments but apparently not on edits made to existing comments.
>
> To point 1 -- We apologize but we do not comprehend the concern. Please help us understand what is the exact issue raised.
>
> A positive pair occurs when the masked part matches the rest of the vector. By way of analogy, the key matches the lock. That $b_i^3$ and $a_i^6$ overlap seems irrelevant to us, since positive pairs are complementary, not overlapping (the missing “part” for $b_i^3$ is $a_i^3$).
>
> To point 2 -- We understand that it is not intuitive to match two vectors of very different sizes. However, the matching function is asymmetric.
>
> In word2vec’s CBOW architecture one predicts a single word from a window of surrounding context words. In BERT one predicts 15% of the tokens based on 85% of the tokens. Our method predicts (more precisely, selects out of $d-k+1$ options) a vector of length $k$ based on a vector of $d-k$. In the extreme case $k=1$. This choice of parameter is not optimal but is still very effective.
>
> Can you please point out the places in which the above analogy with word2vec fails?
>
> As a side note:  In the continuous skip-gram variant of word2vec, the model uses a single word to predict the surrounding window of context words. In this case, the success of word2vec may be even less intuitive according to the argument that $d-k$ and $k$ should be similar in length.
>
> Thank you

---

> > ### Comment · Reviewer_e9Pz · 2021-11-27
> > **Thank you for the responses.**
> >
> > (1) I understood this point now.
> > - F and G are different functions; thus, even though x and y are highly overlapped, F(x) and G(y) can be different.
> >
> > (2) It is somewhat different from word2vec.
> > - Here, if k=1, it tries to match "one" scalar information to "k-1" scalar features which is actually not that intuitive.
> > - I understand that k=1 is not the optimal hyper-parameter; however, the experiments show very promising results even with k=1. Thus, I just want to know what is the real source of gains for this method.
> >
> > I had a misunderstanding on the first point (negative pairing). Thus, I increase the score form 5 to 6.

---

> > > ### Author Response · Authors · 2021-11-28
> > > **Thank you for updating the review**
> > >
> > > Regarding the k=1 intuition. We agree that this goes against the intuition that vectors are informative and scalars are contextless. However, the proposed method matches a context with the scalar, so there is a context to match the scalar against. Intuitively, this is not outside the realm of possibility since, for example, imputing a missing element is possible, and since other lines of work, e.g., in NLP, show that this type of learning works.
> > >
> > > Maybe the following would provide more data to ground this success: in the new experiments requested by iWxT, we apply k=1 as a preprocessing step on the normal-class data to identify noise features. As we show in the plots, one can easily predict the correct match in the k=1 case if the features are not noise features. As expected, for noise features, no prediction is possible. In fact, a threshold is set at ten times of the best prediction value seems to work perfectly on all datasets. This means that all predictions of real k=1 features are within ten times the NLL of the best NLL, and none of the noise features are. It is rare to see such a clear separation.

---

### Official Review · Reviewer_uGiC · 2021-11-01

**Correctness:** 4
**Technical Novelty And Significance:** 3
**Empirical Novelty And Significance:** 4
**Recommendation:** 8
**Confidence:** 4

**Main Review:**

Strengths:
- Interesting use of contrastive learning for anomaly detection on tabular data
- Strong empirical performance outperforming recent baselines; extensive comparisons on ~30 datasets
- Ablation studies and sensitivity analyses for hyperparameters are performed
- Paper is well written and easy to follow; limitations and assumptions are discussed
- Code is provided for reproducibility

Weaknesses:
- My major concern is that this work is very similar to the recent ICML 2021 paper "Neural Transformation Learning for Deep Anomaly Detection Beyond Images", which aims to learn the appropriate masking strategy for tabular data; this is in some sense a more general formulation compared to the fixed masking strategy used for contrastive learning here. Comparison and discussion of this work should be included though it can be to some extent be considered contemporaneous, and does not perform as extensive comparisons on tabular data.
- It is a bit unsatisfying that the self-supervised task is solvable exactly without the need for the learned neural projections yet the success of the method relies on such neural projections; to be fair the paper does include some discussion of this point. Perhaps a more thorough exploration of the need for capacity constraints on the neural network will provide more insight into the method; one could experiment with different network architectures (e.g depth) and/or investigate how anomaly detection performance varies with overlap classification performance.

Other comments/suggestions:
- It may be easier to compare the methods in Table 2 if the ranks and average ranks of the methods are provided.
- Typos:
  * In Table 3 caption: resutls -> results
  * In and around Eq (1) - $\Phi$ instead of $\phi$


**Summary Of The Paper:**

This paper introduces an anomaly detection algorithm for tabular data based on contrastive learning in the semi-supervised setting (training set assumed to be only normal data). The contrastive learning task is based on masking: the features from a single training example are split into two groups (pairs), one being a subsequence of the features, and the other its complement set; the learning task is then to differentiate pairs of splits that overlap (positive) vs not (negative) (from the same split or not) for each sample. The contrastive loss defined by this task is used to train the model and as the anomaly score. The proposed method is extensively evaluated on a range of tabular datasets where it outperforms recent methods.

**Summary Of The Review:**

This is a paper presenting an interesting application of contrastive learning for anomaly detection on tabular data, with extensive experiments showing the strong empirical performance of the method. Ablation studies and sensitivity analyses are provided, as is the code for reproducibility. The main issue with this paper is the similarity to the somewhat contemporaneous ICML 2021 paper listed above, which should be discussed and compared to. On balance the relative novelty and strength of the experiments lean me towards acceptance.

**Post Response Update:**
The authors have included a discussion of the related ICML 2021 paper and highlighted the differences, as well as added additional comparisons to baselines that support the superiority of the proposed method. I think the paper is a clear accept now.

---

> ### Author Response · Authors · 2021-11-16
> **We appreciate the supportive review and the constructive comments.**
>
> We appreciate the supportive review and the constructive comments.
> Thank you for pointing us to the contemporaneous work "Neural Transformation Learning for Deep Anomaly Detection Beyond Images" (the method NeuTraL AD), which we were not aware of. The existence of this work points to the timeliness of our method.
> The two methods are very different. The major differences are:
> 1. NeuTraL AD learns specific masks, while we apply the entire set of the masks specified by a window size $k$.
> 2. The role of NeuTraL AD masks is to mask-out parts that are irrelevant for specific classes. In our case, we perform a two-sided matching that identifies the masked part from the original.
> 3. NeuTraL AD learns a single feature extractor (“encoder”) for both original and transformed data. In our case, the two sides of the contrastive loss are of very different dimensions ($d-k$ and $k$) and we employ two different encoders. Furthermore, we explore in our ablation (and theoretically motivate) the importance of using encoders with different architectures.
> 4. Their method is built such that there is diversity between the different views and similarity between each view and the original. This requires the views to be spread, in the latent space, around the original sample. In our case, the constraints are such that the crop matches the complement elements of the vector better than all other crops. There is no requirement that other crops be dissimilar among themselves.
> 5. Our method, due to its architecture, is equipped with a natural interpretability method for each decision. Their method provides a global mask that discriminates between important and less important features.
> This is now added to the paper.
>
> Regarding the deterministic solution to the self-supervised task, we wish to convince the readers that this is an eye-opening phenomenon, which teaches us about the nature of self-supervised representation learning. As detailed in the paper (Sec. 5), many of the most successful self-supervised methods, which are based on image transformations, can be solved by a simple deterministic computer vision algorithm. This program would, given two images, decide if there is a transformation that links them. The fact that despite this reduction to a simple image-level solution one can learn very powerful semantic representation by using networks teaches us much  about the role of capacity and inductive bias as ways of obtaining meaning. In other words, semantic information can emerge from the data more efficiently than “simpler” low-level solutions. If we may be overly enthusiastic: this is a non-intuitive insight about the nature of representation learning.
>
> As for the request to add alternative architectures, this was done in what was Tab. 7 of the submitted manuscript. We have now added Appendix F of the revision for a discussion of these results. As predicted by the reviewer, adding more layers, which may allow the network to learn the generic solution, does hurt performance. To understand this better, we have added experiments to support the hypothesis that this is caused by the weakening of assumption 2 when extra capacity is added.
>
> Following the review, we added to Appendix A a graph comparing anomaly detection performance (F1 score, x-axis) with classification performance (log-likelihood, assumption 1) and with the classification performance gap between classes (assumption 2). One can observe that the first assumption holds almost globally and that the F1 score is correlated with the difference that is used to quantify the second assumption.
>
> At the request of the reviewer, we have added the mean ranks of the methods to Tab. 2 (adding all ranks would require adding a large table and this information can be inferred from existing data).
> The typos mentioned have been corrected in the revised version. Thank you!

---

### Official Review · Reviewer_iWxT · 2021-11-02

**Correctness:** 4
**Technical Novelty And Significance:** 3
**Empirical Novelty And Significance:** 3
**Recommendation:** 8
**Confidence:** 4

**Main Review:**

I am very surprised this method performs so well, since this k-sliding window does not seem to make sense for tabular data since heterogeneous types of features exist (e.g. discrete, continuous or skewed distribution) and the order of features should not matter and thus the sliding-window approach should not work unlike images. But my concern is much relieved after seeing the complete set of experiments and the stability analysis, as well as some limitations the authors admit in the discussion.

My concerns are as follows (ordered by high to low concerns):
1. Since it's in the tabular data, I hope the authors can compare with long-lasted traditional baselines such as RRCF and KNN in ODDS benchmark (authors can use the default hyperparameter in the sklearn).
2. The claimed that default set of hyperparameters achieve good performance is a bit exaggerated. A better way to phrase is a default "rule" of hyperparameter setting. Because this method still tunes hyperparameters but based on some rules-of-thumbs like the model stops when loss < 10^{-3} when d < 40 else stops when loss < 0.01, or k has to set to d - 150 when k > 150 etc. Overall I still like the stability of this method.
3. I am wondering if authors try adding the location index into the inputs of the model? This might help improve the model to learn better representation based on index j without learning independent model for each j.
4. The Dolan-More profile should be described more clearly. The text is quite confusing right now.
5. In the result of Arrhythmia, the claim is the best F1 score is 63 but all values are between 58.59% to 67.3%. Isn't the best score 67.3?



**Summary Of The Paper:**

## After rebuttal
I am impressed by the comprehensive experiments and baselines and increase my score. Authors provide very complete baselines and also compare in other datasets, which greatly reduces the concern that the proposed rule overfits to the ODDS benchmark.


## Summary
This paper learns a contrastive representation that helps differentiate between normal and abnormal data in the tabular data. Contrary to other contrastive learning methods which learns to differentiate between different examples, this paper instead differentiates between in-window vector and out-window vector for each example under a sliding k-sized window (k is a hyperparameter). Since the features are unordered, they shuffle the feature orders to get multiple score and average them. They show that it outperforms other baselines in a large suite of tabular data, ODDS benchmark, with default hyperparameter rules (e.g. the k is set based on number of samples N and d). They compare mainly with recent methods DROCC, GOAD and COPOD.

**Summary Of The Review:**

Pros:
+ Good rule of hyperparameters which helps reproduce this method.
+ Complete set of experiments and ablation study for the architecture choices.
+ Good stability study of the sliding window k
+ Not sure how authors find out the effectiveness of tanh and two normalization schemes, but they seem to perform well.

Cons:
- No comparison to long-lasted baselines such as RRCF and KNN in the ODDS.
- I think the claim should be changed to "a default rule of hyperparameter selections performs well".

I would put the marginal acceptance for now but I will be happy to raise my score if the baselines are provided in the ODDS.

---

> ### Author Response · Authors · 2021-11-16
> **Thank you very much for the supportive and comprehensive review.**
>
> Thank you very much for the supportive and comprehensive review. We have no factual dispute with it.
>
> For clarification: the method does not rely on using consecutive features. It is only a matter of implementation that we use consecutive features. As we show, results are very similar when the features are permuted and other subsets of features are taken. Our conclusion is that “the original order [i.e., consecutive features] does not provide a clear advantage over a random order of features”. These experiments are detailed in the 2nd paragraph on page 8 (submission version).
>
> Calling the use of consecutive features “a soft assumption” instead of “a straightforward implementation choice” seems to be confusing, and in the revised version (end of Sec. 1) we add wording to clear the confusion. As noted above, this implementation choice does not seem to be preferable to the alternative of selecting random non-overlapping subsets.
>
> 1. We have added the requested baselines to the revised version. Our method still leads but KNN is a surprisingly strong baseline.
>
> 2. We accept the need to restate the wording about a default set vs. default rule. We appreciate the statement about the stability of the method to its parameters. Our method is very stable to the choice of parameters, and together with the very sizable performance gap over the previous methods, we can use an almost arbitrary set of fixed parameters and still show markedly better accuracy than the existing methods.
>
> 3. We experimented with positional embedding, but were unable to achieve notable improvement, if any. Of course, further efforts to condition one of the networks on the location index may improve the method.
>
> 4. In the revised version we made an effort to describe the Dolan-More profile more clearly (Fig. 2 and Sec. 4). The Dolan-More profile is a powerful comparison tool, and we would be happy to see it adopted more widely.
>
> 5. The values in the 58.59% to 67.2% range were obtained by experimenting with different k values. 61.8% is the score obtained by using the default setting for k (which is 63 in this case), and thus this is the result we reported (assuming one cannot optimize the hyper-parameters in this unsupervised setting).

---

> > ### Comment · Reviewer_iWxT · 2021-11-22
> > **Thank you for providing other baselines**
> >
> > Thank you for providing other baselines. I am a little bit surprised to see that kNN performs so well in this benchmark. Since kNN performs much worse when facing high dimensionality or noisy features (shown in the PIDForest paper), it seems that this benchmark is probably small and have very few noisy features. I also think your method will probably not perform well when there are noisy features, since learning the representation between noisy features is also not meaningful. I believe including this point in the limitation will better help readers understand.
> >
> > I also agree with the reviewer e9Pz of the point 4 that the proposed rule for hyperparameters could overfit too much on this benchmark, since these rules are probably tuned based on the test performance on this benchmark. And I feel the authors' response does not fully address this concern.
> >
> > But overall I still think this paper proposes an interesting method and provide a complete study on this benchmark, and still vote for marginal acceptance.

---

> > > ### Author Response · Authors · 2021-11-23
> > > **Thank you for challenging us!**
> > >
> > > 1. Thanks for pointing out a possible limitation when dealing with data that contains noisy features. In the experiment presented in Table 3 of PIDForest, 50 random noisy features are added to the data, which does reduce performance for all methods. PIDForest, as a method that relies on multiple random samples of the features, can handle such noise relatively well.
> > > Our method can very easily detect noisy features, without relying on sampling subsets of the features. The key factor is that random features cannot be predicted from the rest of the features. This can be exploited in the following simple procedure:
> > > a) Run our method with k=1 using cross-validation on the train set.
> > > b) Eliminate all features for which the NLL is larger than 10x the smallest NLL, aggregated across the cross-validation splits (the separation between noisy features and the rest is extremely clear, so almost any threshold would work (see https://imgur.com/a/sL71FCY).
> > > c) Rerun our method with the default k value.
> > >
> > > As can be seen in the table below, our method can handle 50 random features, as in the PIDForest, as well as 200 such features, which is more than what PIDForest can handle. KNN, as shown by PIDForest, and as mentioned by the reviewer, is not competitive with the addition of these noisy features.
> > >
> > > | Thyroid | 0 | 50 | 100 | 200 |
> > > |-----------|----|-----|-----|-----|
> > > | PIDForest | 84.7% | 70.1% | 66.0% | 50.3% |
> > > | KNN       | 38.8% | 11.8% | 6.1% | 6.1% |
> > > | Our*       | 76.8% | 76.8% | 76.8% | 76.8% |
> > >
> > > | Mammography | 0 | 50 | 100 | 200 |
> > > |-----------|----|-----|-----|-----|
> > > | PIDForest | 24.2% | 19.5% | 18.5% | 19.2% |
> > > | KNN       | 38.8% | 6.3% | 5.1% | 6.3% |
> > > | Our*       | 29.4% | 29.4% | 29.4% | 29.4% |
> > >
> > > *Our results remain equal despite the addition of the noisy features due to the perfect detection of these features.
> > >
> > > 2. We believe that we have provided the most extensive set of experiments in any anomaly detection paper. However, following your comment on KNN being successful on ODDS since it may contain small datasets, we have searched google for “large anomaly detection datasets” and found the “ADRepository” benchmark (https://github.com/GuansongPang/ADRepository-Anomaly-detection-datasets).
> > > We ran all the top-performing baseline methods from our paper on all of the numerical datasets included in the repository.
> > > As can be seen, our method leads over the most effective baselines by a large margin.
> > > These results also attest to the generality of our hyperparameter setting rule, since no adjustment was made.
> > >
> > > | Method | mean F1 |
> > > |---------|-------|
> > > | Copod   | 30.9% |
> > > | IForest | 24.5% |
> > > | PID     | 19.2% |
> > > | KNN     | 26.8% |
> > > | GOAD    | 33.1% |
> > > | Ours    | 46.6% |
> > >
> > > 3. We are sorry that you found our reply regarding the hyperparameter setting only partly satisfying. We would like to mention that this rule simply selects between k=2, k=10, or k=d-150 in order not to use a small k in large dimensions. As can be seen in Fig. 7 and 8, our method is very much stable with respect to k. Moreover, as we show in Tab. 7, by simply selecting k=1, the performance in comparison to the hyperparameter setting rule drops modestly, and our method remains better than the baseline methods.

---

> > > > ### Comment · Reviewer_iWxT · 2021-11-23
> > > > **Thank you for very complete sets of experiments; Score increases**
> > > >
> > > > I am not sure authors are still able to comment at this stage. But just leave the questions I have here:
> > > >
> > > > - Q1: If you eliminate the features that bigger than 10 times likelihood, will your result still the same in the ODDS benchmark?
> > > > - Q2: Do you run all the datasets in ADRepo? How is your performance compared to DevNet? Overall it looks great.

---

> > > > > ### Author Response · Authors · 2021-11-23
> > > > > **Thank you for your support**
> > > > >
> > > > > 1. From the unmodified datasets we checked so far it seems that running the filtering procedure will not eliminate any feature. Attached (https://imgur.com/a/HXZSj4J, https://imgur.com/a/5gUKuob, https://imgur.com/a/DTwuXux) are figures depicting the nll on the original ‘Pima’ and ‘Wine’ datasets (without any added noise features) when following the noise detection procedure we mentioned in the previous comment. As can be seen, the nll on all features is very low and consistent.
> > > > > 2. The ADRepository includes four categories of datasets: numerical, categorical, video and images. The results reported are obtained by evaluating the 7 datasets included in the ‘numerical’ category.
> > > > > 3. The DevNet method assumes there is access to labeled anomalous instances in the training phase, and leverages that to learn about the distribution of anomalies. Our method works in a completely unsupervised setting, thus the methods are uncomparable.

---

### Official Review · Reviewer_yptt · 2021-11-09

**Correctness:** 3
**Technical Novelty And Significance:** 2
**Empirical Novelty And Significance:** 3
**Recommendation:** 6
**Confidence:** 4

**Main Review:**

*Pros*
+ Anomaly detection on general multivariate data (where few prior knowledge is available) is an important problem with many applications (fraud detection, etc.) that is relevant to the community.
+ The proposed method performs favorably over recent state-of-the-art methods (DROCC, GOAD, COPOD) on many datasets. Overall, the experimental evaluation is extensive (4 + 30 datasets), scientifically rigorous (including Wilcoxon signed rank tests for best vs. second best performing method), and includes sensitivity analyses for critical method hyperparameters.
+ The paper is overall structured well and easy to follow.
+ The work is placed well into the existing literature.

*Cons*
- The methodological novelty is rather low (contrastive learning for anomaly detection has been recently explored [1, 2], though limited to image data).
- There is not much of an explanation or intuition as to *why* the proposed method seems to work well.


*Questions*
(1) Could you elaborate more on why simply taking consecutive features seems to work well, although tabular data is generally permutation invariant? I appreciate the empirical evidence provided in Appendix A, but I struggle to understand the underlying reason for why this simple heuristic seems to work well on many datasets.


*References*
[1] K. Sohn, C.-L. Li, J. Yoon, M. Jin, and T. Pfister. Learning and evaluating representations for deep one-class classification. ICLR, 2021.
[2] J. Tack, S. Mo, J. Jeong, and J. Shin. CSI: Novelty detection via contrastive learning on distributionally shifted instances. NeurIPS, 2020.

**Summary Of The Paper:**

This paper proposes a contrastive learning method for unsupervised anomaly detection on general multivariate (tabular) data. The idea of the method is to train two encoders, one that embeds a subset of k features and one that embeds the residual d − k features, such that the resulting embeddings are closely aligned via the contrastive loss, where all other subsets of k features are taken as negative examples respectively. This is then repeated over all subsets of features of size k. Thus, the intuition of the approach is to extract common dependencies (e.g., correlations, statistical redundancy, etc.) between the features by achieving a small contrastive loss over the training data, thereby obtaining a high contrastive loss for anomalies (which presumably lack these common dependencies). An empirical evaluation on the Arrhythmia, Thyroid, KDD, and KDDRev datasets (where results from the literature exist) as well as 30 additional datasets from the ODDS library show that the proposed method performs favorably over other recent competing methods (DROCC, GOAD, COPOD) on tabular data.

**Summary Of The Review:**

I think this work presents a simple method that shows significant improvements over previous methods in an extensive and scientifically rigorous evaluation, which makes this work a solid contribution to an important problem (anomaly detection on multivariate, tabular data). For this reason, I am leaning towards accepting this work, though I have some remaining questions and concerns (see above).

---

> ### Author Response · Authors · 2021-11-16
> **Thank you very much for your supportive and comprehensive review.**
>
> Thank you very much for your supportive and comprehensive review. We have no factual dispute with it.
>
> As you mention, previous work on contrastive learning (cited at the end of Sec 2) is indeed inappropriate for image data. As we wrote: “These image transformation-based techniques are not applicable to tabular data, since there is no group of transformations that the content of generic vectors is invariant to.”. We, therefore, had to come up with a different scheme.
>
> Even if our approach looks simple in hindsight, we believe that coming up with the specific approach required a significant inventive step. For example, as reviewer iWxT writes: “contrary to other contrastive learning methods which learn to differentiate between different examples, this paper instead differentiates between in-window vector and out-window vector for each example under a sliding k-sized window.”
>
> As to explaining why the approach works: it relies on two assumptions (top of page 2). If these assumptions are met, the learning problem that is used follows naturally, since, unlike many other anomaly detection methods, the training objective and the test score are exactly the same. The two assumptions are simple to state and easy to validate, as shown in Appendix A.
>
> As to your question: the method does not rely on using consecutive features and the use of such features is only a matter of implementation. As we show, the results are very similar when features are permuted and other subsets of features are taken. Please see the 2nd paragraph on page 8 (submission version). Our conclusion is that “the original order [i.e., consecutive features] does not provide a clear advantage over a random order of features”.
>
> Calling the use of consecutive features “a soft assumption” instead of “a straightforward implementation choice” seems to be confusing, and in the revised version (end of Sec. 1) we add wording to clear the confusion. As noted above, this implementation choice does not seem to be preferable to the alternative of selecting random non-overlapping subsets.

---

### Author Response · Authors · 2021-11-16
**Revised version**

We thank the reviewers for their extremely helpful feedback. We are uploading a revised version of the manuscript. The major modifications are:

[1] The addition of the contemporaneous work by Qiu et al. to the related work section, including a detailed comparison between the methods.

[2] The addition of additional baselines requested by the reviewers.

[3] An additional quantitative analysis to Appendix A, further validating our assumptions.

[4] A discussion of the effect of changing the architecture in the new Appendix F.

[5] The wording has been modified in multiple places, based on the reviewers’ requests and comments.

We once again thank the community for helping us improve our manuscript and welcome additional questions and concerns.

---

### Public Comment · ~Elena_Burceanu1 · 2022-06-22
**Question for the authors**

Hello there, interesting work. I have a question on how the G networks are trained. I see in the paper that "The same networks F, G are learned for all samples i of the training set and for all starting indices j".

Does this mean that the G network receives as input let's say: 1+2+3 features, but also 2+3+4 for instance, but in the same position like the first combination, in the same training iteration? If so, is this ok, what does G learn in the end in this scenario, if the feature position does not matter? Am I getting this wrong (As I understand it, this sounds like getting for instance some "age" feature in the first position, in the next sample in the second one, and so on)?

Thank you for your answer,
Elena

---

> ### Public Comment · ~Lior_Wolf1 · 2022-06-22
> **Thank you for your question!**
>
> You understood correctly, Elane.
>
> It is the same network. There is only one network and no positional embedding. While one can use multiple networks or positional embedding, we opted for a simple and efficient solution.
>
> The networks need to learn to classify regardless of the location of the features. There seems to be enough capacity to do so.
>
> If you try variants of the method with multiple network pairs, we'd be happy to hear how it turned out.
>
> Thanks

---

> > ### Public Comment · ~Elena_Burceanu1 · 2022-06-22
> > **Thank you for your fast answer**
> >
> > This sounds a bit counter-intuitive since the problem ("to classify regardless of the location of the features") might not be well defined, and it is hard for me to understand what G learns in the end (except for some mean value over the output). So I question if this is rather a problem of the network capacity or not.
> >
> > For instance, for the tabular case where the features mean different things, we have some blood test results. These features get the same position (e.g. first), but meaning a different thing, and would only distract the network. I see the case of missing positional encodings for words or patches a bit different, because at least the features mean the same thing. But I presume you wanted to show that even if the G network is extremely basic (and learns maybe only some kind of bias), this approach still works.
> >
> > Thanks for the clarifications,
> > Elena

---

> > > ### Public Comment · ~Lior_Wolf1 · 2022-06-22
> > > **Thank you for the follow-up question**
> > >
> > > I would say that in the way we did it, the network has the additional task of identifying the location. In your scenario, this additional task is easier than in uniform cases.
> > >
> > > Our experiments imply that the networks have enough capacity to cover all locations. I would be very happy to hear your results if you try to run networks with less capacity that are specialized for different locations.
> > >
> > > Thanks again,
> > >
> > > Lior

---

### Public Comment · ~Bo_Qin1 · 2022-11-28
**Question for the authors' code**

Hello there, I got a question about the code for loading the KDD dataset in your "build_train_test_kdd" function, which I posted below.
You seem to assign the normal data as 1.0 while the abnormal data as 0.0 in the 41st column for each row. However, kdd_normal, which is the variable that you want to store the normal data, is assigned by finding the rows where "revised_pd[41] == 0.0" which is the anomaly data. So why was that? Did you make a mistake?

Thanks for your reply.
```python
    def build_train_test_kdd(self, name_of_file):
        # some unrelated stuff
        revised_pd = revised_pd.reindex(columns=new_columns)
        revised_pd.loc[revised_pd[41] != 'normal.', 41] = 0.0
        revised_pd.loc[revised_pd[41] == 'normal.', 41] = 1.0
        kdd_normal = np.array(
            revised_pd.loc[revised_pd[41] == 0.0], dtype=np.double)
        kdd_anomaly = np.array(
            revised_pd.loc[revised_pd[41] == 1.0], dtype=np.double)
        kdd_normal = torch.tensor(kdd_normal)
        kdd_anomaly = torch.tensor(kdd_anomaly)
        kdd_normal = kdd_normal[
            torch.randperm(kdd_normal.shape[0])]
        kdd_anomaly = kdd_anomaly[torch.randperm(kdd_anomaly.shape[0])]
        train, test_norm = torch.split(
            kdd_normal, int(kdd_normal.shape[0] / 2) + 1)
        test = torch.cat((test_norm, kdd_anomaly))
        test = test[torch.randperm(test.shape[0])]
        test_classes = test[:, -1].view(-1, 1)
        train = train[:, 0:train.shape[1] - 1]
        test = test[:, 0:test.shape[1] - 1]
        return (train, test, test_classes)
```

---

> ### Public Comment · ~Lior_Wolf1 · 2022-11-28
> **Thank you for your question!**
>
> I believe that we treated the dataset in the same manner Bergman and Hoshen ICLR 2022 (and other previous publications) did.
> This is from that work:
> >Following Zong et al. (2018), we evaluate two different settings for the KDD dataset:
> KDDCUP99: In this configuration, we use the entire UCI 10% dataset. As the non-attack class consists of only 20% of the dataset, it is treated as the anomaly in this case, while attacks are treated as normal.
> KDDCUP99-Rev: To better correspond to the actual use-case, in which the non-attack scenario is normal and attacks are anomalous, Zong et al. (2018) also evaluate on the reverse configuration, in which the attack data is sub-sampled to consist of 25% of the number of non-attack samples. The attack data is in this case designated as anomalous (the reverse of the KDDCUP99 dataset)

---

### Public Comment · ~Peigen_Ye1 · 2023-05-18
**About the F1-score in your code**

I read your paper and learned a lot from it. When I looked at the code, there was one thing that confused me. About the 'f1_calculator' in helper_functions.py, I found that you calculated the F1 score using 'true_pos / (true_pos + false_pos)'. But, as far as I know, that's the way to calculate 'precision'. I'd appreciate it if you could explain it to me.

```
def f1_calculator(classes, losses):
    classes=classes.numpy()
    losses=losses.numpy()
    df_version_classes = pd.DataFrame(data=classes)
    df_version_losses = pd.DataFrame(losses).astype(np.float64)
    Na = df_version_classes[df_version_classes.iloc[:, 0] == 1].shape[0]
    anomaly_indices = df_version_losses.nlargest(Na, 0).index.values
    picked_anomalies = df_version_classes.iloc[anomaly_indices]
    true_pos = picked_anomalies[picked_anomalies.iloc[:, 0] == 1].shape[0]
    false_pos = picked_anomalies[picked_anomalies.iloc[:, 0] == 0].shape[0]
    f1 = true_pos / (true_pos + false_pos)
    return (f1)
```

---

> ### Public Comment · ~Tom_Shenkar1 · 2023-05-23
> **Thank you for asking**
>
> As we noted in the paper, we followed the protocol of Zong et al. (2018) and Bergman & Hoshen (2020), according to it the threshold for determining the amount of predicted anomalies is set by the actual number of anomalies in the dataset. Hence the amount of false-positives is equal to the amount of false-negatives. Thus in this case the F1 score is the same as the precision.
> Hope this clarifies it.

---

### Decision · Program_Chairs · 2022-01-20

**Decision:**

Accept (Poster)

**Comment:**

The paper proposes contrastive learning for tabular data to improve anomaly detection.
Strengths:
- Interesting and important problem.
- Usage of contrastive learning for anomaly detection in general multi-variate datasets is novel (as prior work mostly focuses on images)
- Extensive experiments with comparisons to multiple baselines on multiple datasets
- Well-written paper

The reviewers raised some concerns about novelty (in particular, the relationship to the closely related paper "Neural Transformation Learning for Deep Anomaly Detection Beyond Images"), hyperparameter tuning and additional baselines. The authors did a great job of addressing the concerns and multiple reviewers raised their scores. During the discussion phase, the consensus decision leaned towards accept. I recommend acceptance and encourage the authors to address any remaining concerns in the final version.

Additional AC comments:
- Please make sure that the camera ready version does not exceed page limits. https://iclr.cc/Conferences/2022/CallForPapers
- " As far as we can ascertain, masking was not used for one-class classification before.": There's some related work on pre-training BERT for OOD detection (cf. https://arxiv.org/abs/2004.06100 or https://arxiv.org/abs/2106.03004) which might be worth discussing.